# Towards Unbounded Machine Unlearning

**Meghdad Kurmanji**[*]  
University of Warwick

**Peter Triantafillou**  
University of Warwick

**Jamie Hayes**  
Google DeepMind

**Eleni Triantafillou**  
Google DeepMind

## Abstract

Deep machine unlearning is the problem of 'removing' from a trained neural network a subset of its training set. This problem is very timely and has many applications, including the key tasks of removing biases (RB), resolving confusion (RC) (caused by mislabelled data in trained models), as well as allowing users to exercise their 'right to be forgotten' to protect User Privacy (UP). This paper is the first, to our knowledge, to study unlearning for different applications (RB, RC, UP), with the view that each has its own desiderata, definitions for 'forgetting' and associated metrics for forget quality. For UP, we propose a novel adaptation of a strong Membership Inference Attack for unlearning. We also propose SCRUB, a novel unlearning algorithm, which is the only method that is consistently a top performer for forget quality across the different application-dependent metrics for RB, RC, and UP. At the same time, SCRUB is also consistently a top performer on metrics that measure model utility (i.e. accuracy on retained data and generalization), and is more efficient than previous work. The above are substantiated through a comprehensive empirical evaluation against previous state-of-the-art.

## 1 Introduction

Can we make a deep learning model 'forget' a subset of its training data? Aside from being scientifically interesting, achieving this goal of deep machine unlearning is increasingly important and relevant from a practical perspective and this research direction is enjoying significant attention recently [Triantafillou et al., 2023]. Regulations such as EU's General Data Protection Regulation [Mantelero, 2013] stipulate that individuals can request to have their data 'deleted' (the 'right to be forgotten'). Nowadays, given the ubiquity of deep learning systems in a wide range of applications, including computer vision, natural language processing, speech recognition and healthcare, allowing individuals to exercise this right necessitates new technology. Unlearning aims to fill this need.

However, unlearning has several additional important applications where privacy with respect to a subset of individual training examples isn't necessarily a consideration: removing out-of-date examples, outliers, poisoned samples [Jagielski et al., 2018], noisy labels [Northcutt et al., 2021], or data that may carry harmful biases [Fabbrizzi et al., 2022]. In fact, we are the first to argue that the definition of 'forgetting' is application-dependent, and unlearning comes with different desiderata and priorities in different scenarios. For instance, for a privacy application, 'forgetting' the data of users who requested deletion is the primary objective and we may be willing to sacrifice model performance to some degree, in exchange for reliably defending Membership Inference Attacks (MIAs). In other cases, for instance, when deleting outdated data to keep a model current, maintaining the model's performance throughout such 'cleanup' operations may be the primary consideration, and guaranteeing that no trace of the old data is left behind may be irrelevant. While previous work does not establish such distinctions, we study three scenarios and propose a method that is the most consistent top-performer across applications and metrics.

---

[*]Correspondence to `Meghdad.Kurmanji@warwick.ac.uk`.

37th Conference on Neural Information Processing Systems (NeurIPS 2023).

Unfortunately, removing the influence of a subset of the training set from the weights of a trained model is hard, since deep models memorize information about specific instances [Zhang et al., 2020, 2021, Arpit et al., 2017] and their highly non-convex nature makes it difficult to trace the effect of each example on the model's weights. Of course, we can apply the naive solution of retraining the model from scratch without the cohort of data to be forgotten. This procedure indeed guarantees that the weights of the resulting model aren't influenced by the instances to forget (it performs '*exact unlearning*'), but the obvious drawback is computational inefficiency: retraining a deep learning model to accommodate each new forgetting request isn't viable in practice. To mitigate that cost, recent work turned to *approximate unlearning* [Izzo et al., 2021, Golatkar et al., 2020a,b] where the goal is to modify the weights of the trained model to approximate those that would have been obtained by exact unlearning. That is, they strive to achieve 'indistinguishability' between the approximate and exact solutions, typically accompanied by theoretical guarantees for the quality of that approximation. This goal is also mirrored in the metrics used by these works: the model's error on the deleted data is desired to be just as high as the error of the 'exact' retrained model (which truly never saw that data). The argument is that an error higher than that reference point may cause susceptibility to membership inference attacks: a noticeably poor performance on an example can reveal that it was unlearned.

In this work, we show empirically that such approximate unlearning methods often perform poorly in practice (in one or more of RB, RC and UP). We hypothesize that this is due to potential violations of assumptions made by these methods, e.g. that the 'optimal' solution of unlearning is close in weight space to the original model [Golatkar et al., 2020a,b] (due to assumptions of stability of SGD and 'small enough' forget sets). Further, we find these previous methods scale poorly to the size of the training set and the model, often making them impractical. To address these issues, we propose a new unlearning model, SCalable Remembering and Unlearning unBound (SCRUB) that departs from that methodology. SCRUB bears a novel teacher-student formulation, where the student model selectively disobeys an all-knowing teacher, to inherit from it *only* knowledge that does not pertain to the data to be deleted. This method is 'unbound' from limiting assumptions (e.g., to facilitate formal guarantees), poor scalability limits, and is not tailored to a single unlearning application. We find that SCRUB can yield a very high error on the deleted data, which is desired in some scenarios but may cause susceptibility to membership inference attacks. To mitigate this when relevant, we extend SCRUB with a novel 'rewinding' procedure that pinpoints the 'checkpoint' of the unlearning process to use such that the error on the deleted data approximates the reference point of being 'just high enough'.

We also propose the first, to our knowledge, adaptation of the state-of-the-art LiRA MIA [Carlini et al., 2022] to the framework of unlearning, as a metric for UP. Overall, we perform a comprehensive evaluation using different baselines, datasets, architectures and three application scenarios. We find that SCRUB is by far the most consistent top performer out of all state-of-the-art methods considered. SCRUB is able to ensure high forget quality, satisfying different forget-quality metrics for different applications while also being a top performer with respect to model utility (accuracy on retained data, and generalization). Our findings highlight the need to consider a broad spectrum of applications when evaluating unlearning algorithms, to better understand their properties and trade-offs.

## 2 Problem Definition for Unlearning Unbound

**Notation and preliminaries** Let $\mathcal{D} = \{x_i, y_i\}_{i=1}^{N}$ denote a training dataset containing $N$ examples, where the $i$'th example is represented by a vector of input features $x_i$ and a corresponding class label $y_i$. Let $f(\cdot; w)$ denote a function parameterized by trainable weights $w$. In this work, we study the case where $f$ is represented by a deep neural network, trained in a supervised manner via empirical risk minimization. Specifically, we define the loss function as $\mathcal{L}(w) = \frac{1}{N} \sum_{i=1}^{N} l(f(x_i; w), y_i)$ where $l$ denotes the cross entropy loss.

**Deep Machine Unlearning** We now formalize the problem of deep machine unlearning. We assume that we are given a model $f(\cdot; w^o)$ that has been trained on $\mathcal{D}$, where $f$ denotes a neural network and $w^o$ its trained parameters, obtained by minimizing the above loss function. We will refer to this as the 'original model' (i.e. before any unlearning intervention is performed). We are also given a 'forget set' $\mathcal{D}_f = \{x_i, y_i\}_{i=1}^{N_f} \subset \mathcal{D}$, comprised of $N_f$ examples from $\mathcal{D}$ and a 'retain set' $\mathcal{D}_r$ of $N_r$ training examples that the model is still allowed to retain information for. For simplicity, we consider the standard scenario where $\mathcal{D}_r$ is the complement of $\mathcal{D}_f$ or, in other words, $\mathcal{D}_f \cup \mathcal{D}_r = \mathcal{D}$.

Given $f(\cdot; w^o)$, $\mathcal{D}_f$ and $\mathcal{D}_r$, the goal of deep machine unlearning is to produce a new set of weights $w^u$ such that the 'unlearned model' $f(\cdot; w^u)$ has 'forgotten' $\mathcal{D}_f$ without hurting the 'utility' of the original model (its performance on the retained data $\mathcal{D}_r$ and generalization to data outside of $D$). We argue that defining (and measuring) 'forgetting' is application-dependent. Our work is the first, to the best of our knowledge, to perform a thorough empirical evaluation of unlearning algorithms on three different application scenarios, each with its own definition of forgetting and associated metrics.

**Defining and measuring 'forgetting'** We consider three different application scenarios in this work, each with their own definition of forgetting and associated metrics. 1) **Removing biases**. We want *maximal* error on $\mathcal{D}_f$: the model should *never* predict the assigned labels of forget examples, since they reflect unintended behaviour. 2) **Resolving confusion**. A model may be 'confused' due to incorrect labels in its dataset. A successful unlearning algorithm resolves this if mislabeled examples are placed in the forget set (see metrics in Section 5). 3) **User privacy**. Success from a privacy perspective is associated with a forget error *only as high as that of retraining-from-scratch*; a higher forget error could lead to vulnerability to Membership Inference Attacks (MIAs). In all of the above cases, in addition to achieving forgetting, we seek unlearning algorithms that don't damage *model utility*, i.e. accuracy on retained data and generalization ability, and that unlearning is *efficient* enough to be preferable over the simple and expensive solution of retraining the model.

**Fundamental trade-offs** A top-performing unlearning algorithm must simultaneously achieve three feats: (i) obtain high 'forget quality', (ii) ensure continued high accuracy on retained and unseen data, and (iii) efficiency / scalability. In practice, there are fundamental trade-offs in two different dimensions. The first dimension involves trade-offs between (i), (ii) and (iii), for example achieving good 'forget quality' at the expense of efficiency. Retraining from scratch is such an example, but, as we will show later, some methods suffer from inefficiencies even worse than retraining. Another common issue we will observe is achieving good 'forget quality' at the expense of utility. The second dimension involves trade-offs *within* the context of achieving good 'forget quality' across applications, since the metric for this quality is application-specific, and an algorithm may perform strongly on one such metric but poorly on others. SCRUB is designed to handle these trade-offs and is shown empirically to consistently outperform previous methods across the board.

## 3 SCalable Remembering and Unlearning unBound (SCRUB)

We present SCRUB, a new unlearning algorithm based on a novel casting of the unlearning problem into a teacher-student framework. We design SCRUB to meet the desiderata of unlearning: efficiently 'forgetting' without hurting model utility. Because we argue that the definition of forgetting is application-dependent, we propose a recipe that works across applications: We first design SCRUB to strive for maximal forget error, which is desirable in some scenarios (e.g. RB and RC) but not in others (e.g. UP). To address the latter case, we complement SCRUB with a novel 'rewinding' procedure that can reduce the forget set error appropriately when required. SCRUB is 'unbound' from limiting assumptions (e.g., to facilitate formal guarantees), poor scalability limits and is by far the most consistent method in forgetting across applications while preserving model utility.

### 3.1 The SCRUB

We consider the original model $w^o$ as the 'teacher' and we formulate our goal as training a 'student' $w^u$ that *selectively* obeys that teacher. Intuitively, our goal for $w^u$ is twofold: forget $\mathcal{D}_f$ while still remembering $\mathcal{D}_r$. To that effect, $w^o$ should be obeyed when teaching about $\mathcal{D}_r$, but disobeyed when teaching about $\mathcal{D}_f$. Our code is available for reproducibility [2].

En route to deriving our training objective, let us first define:

$$d(x; w^u) = D_{\mathrm{KL}}(p(f(x; w^o))\|p(f(x; w^u)))$$

In words, $d(x; w^u)$ is the KL-divergence between the student and teacher output distributions (softmax probabilities) for the example $x$. We make the dependence of $d$ on $w^u$ explicit, since we will optimize the student weights $w^u$ while keeping the teacher weights frozen $w^o$, treating them as a constant.

---

[2] `https://github.com/Meghdad92/SCRUB`

Concretely, we begin by initializing the student $w^u$ to the weights of the teacher $w^o$. Since the teacher weights $w^o$ were trained on all of $\mathcal{D}$, the teacher performs well on both constituents of $\mathcal{D}$, namely both of $\mathcal{D}_r$ and $\mathcal{D}_f$. Therefore, the student has good performance on $\mathcal{D}_r$ at initialization, already fulfilling one of our desiderata. Can we then start from this solution and modify it to unlearn $\mathcal{D}_f$? In principle, one could do this by optimizing:

$$\min_{w^u} -\frac{1}{N_f} \sum_{x_f \in \mathcal{D}_f} d(x_f; w^u) \tag{1}$$

However, in practice, when performing this maximization of the distance between the student and teacher on the forget set, we noticed that, while indeed the performance on $\mathcal{D}_f$ degrades, as desired, it unfortunately also results in degrading performance on $\mathcal{D}_r$. To amend that, we propose to simultaneously encourage the student to 'stay close' to the teacher on retain examples, while encouraging it to 'move away' from the teacher on forget examples, adding a contrastive flavour. Formally, the optimization objective becomes:

$$\min_{w^u} \frac{1}{N_r} \sum_{x_r \in \mathcal{D}_r} d(x_r; w^u) - \frac{1}{N_f} \sum_{x_f \in \mathcal{D}_f} d(x_f; w^u) \tag{2}$$

Furthermore, SCRUB also simultaneously optimizes for the task loss on the retain set, to further strengthen the incentive to perform well there. Our final training objective is then the following:

$$\min_{w^u} \frac{\alpha}{N_r} \sum_{x_r \in \mathcal{D}_r} d(x_r; w^u) + \frac{\gamma}{N_r} \sum_{(x_r, y_r) \in \mathcal{D}_r} l(f(x_r; w^u), y_r) - \frac{1}{N_f} \sum_{x_f \in \mathcal{D}_f} d(x_f; w^u) \tag{3}$$

where $l$ stands for the cross-entropy loss and the $\alpha$ and $\gamma$ are scalars that we treat as hyperparameters.

In practice, we found that optimizing the objective in Equation 3 is challenging, due to oscillations in the loss. Intuitively, this is due to trying to simultaneously satisfy two objectives, which may interfere with each other, namely moving close to the teacher on some data points while moving away from it on others. To address this, SCRUB provides a practical recipe for optimization, reminiscent of 'tricks' used in other min-max-like problems like Generative Adversarial Networks [Goodfellow et al., 2014] where the discriminator is trained for several steps before each update to the generator.

Specifically, SCRUB iterates between performing an epoch of updates on the forget set (the *max-step*) followed by an epoch of updates on the retain set (the *min-step*), in an alternating fashion. Guarding against hurting retain performance due to this alteration of *min-steps* and *max-steps*, SCRUB also performs a sequence of additional *min-steps* at the end of the sequence to restore the retain performance in the event that it was harmed. SCRUB training stops when the forget error has increased without harming the retain set error. We find in practice this point can be reached with a small number of epochs. We provide pseudocode, training plots and ablations in the Appendix.

### 3.2 SCRUB and Rewind (SCRUB+R)

By construction, SCRUB encourages obtaining a high error on the forget set. This is desired for some application scenarios. However, an uncharacteristially high error on deleted examples can make them identifiable, causing vulnerability to membership inference attacks. SCRUB+R addresses this.

To this end, we are after a procedure for selecting the checkpoint of SCRUB where the forget set error is 'just high enough'. One could obtain a reference point for that by retraining from scratch without the forget set and recording the error of that model on the forget set. However, that would defeat the purpose of unlearning, as we want to avoid that computation. Therefore, we propose a different way of establishing a reference point. First, we create a validation set of the same distribution as the forget set. For instance, if the forget set has only examples of class 0, we keep only examples of class 0 in the validation set too. Next, we train SCRUB as usual, storing a model checkpoint every epoch. At the end of its training, where the forget error is typically the highest, we measure the error on the constructed validation set. This will serve as the reference point for the desired forget set error. Finally, we 'rewind' to the checkpoint where the forget error is closest to that reference point.

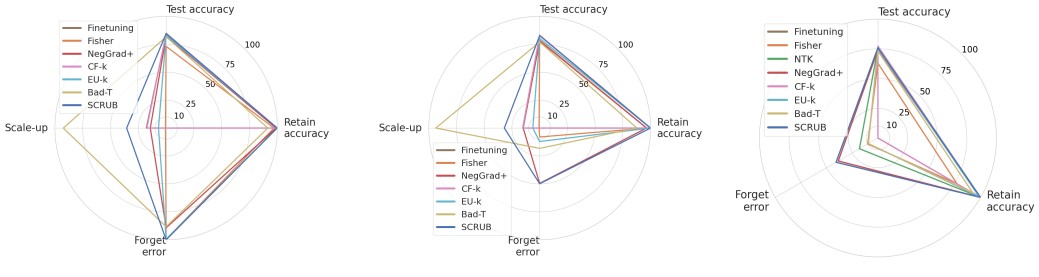

(a) CIFAR-10 class unlearning.  (b) CIFAR-10 selective.  (c) CIFAR-5 selective.

Figure 1: **RB results: SCRUB is the only consistent top-performer in terms of forgetting and preserving utility. It is also highly efficient: second only to Bad-T which, however, fails at forgetting and damages utility**. In all subfigures, each point represents the average across ResNet and All-CNN variants. For large-scale results, we also compute the scale-up factor: the fraction of the runtime of retraining from scratch over the runtime of the given unlearning algorithm (multiplied by 5 here, for visualization purposes). We find that selective unlearning is harder for all methods, especially for EU-k and Fisher (see Figure 1a vs 1b). CF-k and Finetuning perform similarly (they perform poorly) in all cases, so their lines occlude one another. Complete tables are in the Appendix.

The intuition is that the last step of unlearning approximates 'maximally forgetting'. Consequently, the error on the identically-distributed validation set approximates the error of a model that never learned about that distribution from the forget set: any correct predictions on the held-out examples are due only to the generalization power of the model that was trained only on the retain set. Therefore, we choose that validation set error to serve as the reference point for how high we would like the forget set error to be for UP applications. We show empirically that this rewinding procedure greatly increases the defense against membership inference attacks, making SCRUB a strong performer for applications where privacy is a consideration.

## 4 Related Work

**Unlearning definitions**   Despite prior work on unlearning, this research problem is still in its infancy and we lack a well-established formal definition. The term 'unlearning' was coined in Cao and Yang [2015], where, in the context of certain structured problems, unlearning is defined to be successful if deleting an example yields the same outputs on a dataset as if that example had never been inserted. Ginart et al. [2019] introduce a more relaxed probabilistic definition for machine unlearning inspired by Differential Privacy [Dwork et al., 2014] that requires the unlearned model to yield outputs that are *similar* to those that would have been produced by a model retrained from scratch without the forget set. This led to the development of 'approximate learning' methods that aren't identical to retrain-from-scratch ('exact unlearning') but 'similar' [Guo et al., 2019, Golatkar et al., 2020a]. Sekhari et al. [2021] and Neel et al. [2021] adopt similar DP-inspired definitions of (approximate) unlearning based on the goal of indistinguishability from retrain-from-scratch.

However, there is a recent line of criticism against the idea of defining or measuring unlearning success based on indistinguishability from retrain-from-scratch. Thudi et al. [2022b] theoretically show that we can obtain arbitrarily similar model weights from training on two non-overlapping datasets. This implies that reaching a particular point in parameters space isn't a sufficient condition for having unlearned. Furthermore, Goel et al. [2022] argue that it's not a necessary condition either: retraining from scratch may arrive at different model distributions due to changes in e.g. hyperparameters [Yang and Shami, 2020]. Therefore, a unlearning method may be unnecessarily penalized for not matching a *particular* retrained-from-scratch model. In our work, we take a practical viewpoint and define unlearning as 'forgetting' in an efficient manner and without hurting model utility, for three different application-specific metrics for forgetting that we discuss in the next section.

**Unlearning methods**   Cao and Yang [2015] introduce unlearning and provide an exact forgetting algorithm for statistical query learning. [Bourtoule et al., 2021] proposed a training framework by sharding data and creating multiple models, enabling exact unlearning of certain data partitions. Ginart et al. [2019] introduce a probabilistic definition for machine unlearning inspired by Differential

Privacy [Dwork et al., 2014], and formed the origin of the idea of the model indistinguishability goal. More recently, Guo et al. [2019], Izzo et al. [2021], Sekhari et al. [2021] built upon this framework and introduced unlearning methods that can provide theoretical guarantees under certain assumptions. Further, Huang and Canonne [2023] provide bounds for 'deletion capacity' in the DP-based unlearning methods using the group property of DP. [Graves et al., 2021] proposed two methods applicable to deep networks. The first randomly re-labels the forget set and retrains for a small number of updates, while the second stores the index of examples that participated in each batch and the per-batch parameter updates, and unlearns by 'undoing' the affected updates. This requires a lot of storage and can also lead to a poor proxy of the model that would have been obtained had those parameter updates not happened in the first place, especially for larger sequences. [Golatkar et al., 2020a] proposed an information-theoretic procedure for removing information about $\mathcal{D}_f$ from the weights of a neural network and [Golatkar et al., 2020b, 2021] propose methods to approximate the weights that would have been obtained by unlearning via a linearization inspired by NTK theory [Jacot et al., 2018] in the first case, and based on Taylor expansions in the latter. However, [Golatkar et al., 2020a,b] scale poorly with the size of the training dataset, as computing the forgetting step scales quadratically with the number of training samples. [Golatkar et al., 2021] addresses this issue, albeit under a different restrictive assumption: that a large dataset is available for pre-training that will remain 'static', in that no forgetting operations will be applied on it; an assumption that is not met in many important applications (e.g. healthcare, medical images). Some more recent works propose modifications to the original model training, to make the resulting model more amenable to unlearning. Thudi et al. [2022a] propose a regularizer to reduce the 'verification error', which is an approximation to the distance between the unlearned model and a retrained-from-scratch model. The goal is to make unlearning easier in the future. Additionally, they propose a single gradient update that reverses the effect of $\mathcal{D}_f$. This regularizer, however, may have a negative effect on the model's performance, and the single gradient unlearning step only removes the effect that $\mathcal{D}_f$ has had in the first iteration of training. [Zhang et al., 2022] present a training process that quantizes gradients and applies randomized smoothing. This process is designed to make unlearning unnecessary in the future and comes with certifications under some conditions. However, a deletion request that causes too large a change to the data distribution (e.g. if doing class unlearning), may exceed the 'deletion budget' and invalidate their assumptions.

**Related research areas**   Differential Privacy (DP) [Dwork et al., 2014] and Life-Long Learning (LLL) [Parisi et al., 2019] have shared goals with unlearning. DP seeks to ensure that the trained model doesn't store information about *any* training instance; a strong goal that is difficult to achieve for deep neural networks [Abadi et al., 2016]. In contrast, in unlearning we only desire to remove the influence of the training instances in the forget set, which can be seen as a relaxation of the same problem. LLL seeks to continuously update a model to solve new sets of tasks, without 'catastrophically forgetting' previously-learned tasks. The notion of forgetting a task, though, is distinct from that of forgetting a training example: the performance on a task can degrade, without having removed the influence of particular examples from the model weights.

**Teacher-student and contrastive methods**   The teacher-student methodology has been used in a different contexts, e.g. self-supervised [Grill et al., 2020, Chen and He, 2021] and semi-supervised learning [Xie et al., 2020]. Knowledge Distillation [Hinton et al., 2015] can also be thought of as a teacher-student approach with the additional desire for compression, where the student is chosen to have a smaller architecture than the teacher. In addition, our training objective has a contrastive flavour [Chen et al., 2020, He et al., 2020], due to both pulling the student close to the teacher, and pushing it away from it, for different parts of the training set. For unlearning, [Kim and Woo, 2022, Wu et al., 2022] propose methods with a two-stage pipeline: 'neutralization' / forgetting followed by a 'retraining' / restoring, that restores performance that may be lost in the first phase. Both use distillation only in the second phase on only the retain set. In contrast, SCRUB employs a single phase where we carefully design a teacher-student formulation to both encourage forgetting the requested data and retaining the other knowledge, simultaneously. Most similar to our work in that regard is the method of [Chundawat et al., 2022] that employs a related but different teacher-student objective that encourages moving close to a 'Bad Teacher' for the forget set (we thus refer to this as 'Bad-T'). Moving close to a different, 'bad', teacher for forget examples is fundamentally different from our goal of moving away from the (single) teacher as in SCRUB. We find that Bad-T forgets poorly compared to SCRUB in all applications, and degrades model quality.

| Model | ResNet | | | | | | | | All-CNN | | | | | | | |
|---|---|---|---|---|---|---|---|---|---|---|---|---|---|---|---|---|
| | IC test error ($\downarrow$) | | IC retain error ($\downarrow$) | | Fgt test error ($\downarrow$) | | Fgt retain error ($\downarrow$) | | IC test error ($\downarrow$) | | IC retain error ($\downarrow$) | | Fgt test error ($\downarrow$) | | Fgt retain error ($\downarrow$) | |
| | mean | std | mean | std | mean | std | mean | std | mean | std | mean | std | mean | std | mean | std |
| Retrain | 24.0 | 1.8 | 0.0 | 0.0 | 18.3 | 4.2 | 0.0 | 0.0 | 19.0 | 1.3 | 0.0 | 0.0 | 11.3 | 4.6 | 0.0 | 0.0 |
| Original | 56.0 | 3.0 | 0.0 | 0.0 | 92.0 | 7.9 | 0.0 | 0.0 | 49.0 | 4.8 | 6.0 | 10.39 | 80.7 | 12.6 | 6.0 | 10.4 |
| Finetune | 52.0 | 3.1 | 0.0 | 0.0 | 79.3 | 10.1 | 0.0 | 0.0 | 43.7 | 7.3 | 0.0 | 0.0 | 67.3 | 16.0 | 0.0 | 0.0 |
| NegGrad+ | 47.5 | 5.3 | 0.0 | 0.0 | 69.0 | 13.5 | 0.0 | 0.0 | 42.3 | 11.3 | 0.0 | 0.0 | 62.0 | 22.6 | 0.0 | 0.0 |
| CF-k | 54.8 | 2.0 | 0.0 | 0.0 | 85.3 | 7.0 | 0.0 | 0.0 | 47.8 | 5.8 | 0.0 | 0.0 | 76.3 | 14.4 | 0.0 | 0.0 |
| EU-k | 47.0 | 4.8 | 8.3 | 4.7 | 63.3 | 9.7 | 3.7 | 2.5 | 59.5 | 5.2 | 38.3 | 6.7 | 68.7 | 15.6 | 19.7 | 10.4 |
| Fisher | 51.5 | 7.5 | 26.3 | 9.5 | 79.0 | 3.6 | 20.0 | 7.9 | 56.8 | 8.7 | 31.7 | 14.0 | 78.3 | 15.5 | 17.7 | 11.5 |
| NTK | 37.5 | 4.0 | 0.0 | 0.0 | 52.0 | 10.6 | 0.0 | 0.0 | 36.7 | 4.1 | 3.0 | 5.2 | 54.3 | 9.0 | 3.0 | 5.2 |
| Bad-T | 47.3 | 9.7 | 6.0 | 6.2 | **27.0** | 8.7 | 0.3 | 0.6 | **39.3** | 8.2 | 3.0 | 1.0 | **25.7** | 5.0 | 1.0 | 1.0 |
| SCRUB | **19.0** | 3.9 | 0.0 | 0.0 | **19.7** | 7.5 | 0.0 | 0.0 | **26.0** | 4.4 | 0.0 | 0.0 | **18.0** | 11.1 | 0.0 | 0.0 |

Table 1: **RC-application results: SCRUB is the strongest performer in terms of resolving class confusion**. These results are on CIFAR-5 with ResNet and All-CNN, where we confused 50 samples of class 0 by mislabeling them as class 1, and 50 of class 1 by mislabeling them as class 0.

# 5 Experiments

## 5.1 Experimental setup

**Overview of applications and metrics** We consider three sets of forget-quality metrics, for different applications. First, we consider the scenario of **Removing Biases (RB)** where we desire the highest possible forget set error (reflecting that we want the model to *never* predict the associated labels of the forget set, since those capture an unintended behaviour / bias). Next, inspired by Goel et al. [2022], we consider unlearning for **Resolving Confusion (RC)** between classes, caused by a portion of the model's original training set being mislabelled. Finally, we consider a scenario of data deletion for **User Privacy (UP)**. In this case, we use Membership Inference Attacks (MIAs) to measure success (including the first, to our knowledge, adaptation of the state-of-the-art LiRA MIA [Carlini et al., 2022] to unlearning), where unlearning succeeds if the attacker is unable to tell apart forgotten examples from truly unseen ones. In all cases, aside from achieving the application-specific forgetting goal, we also measure utility (performance on $\mathcal{D}_r$ and a test set $\mathcal{D}_t$) as an additional metric.

**Experimental details** We utilize the same two datasets from previous work: CIFAR-10 [Krizhevsky et al., 2009] and Lacuna-10 [Golatkar et al., 2020a], which is derived from VGG-Faces [Cao and Yang, 2015], and the same two architectures: All-CNN [Springenberg et al., 2014] and ResNet-18 [He et al., 2016]. For consistency with previous work, we pre-train the original model on CIFAR-100 / Lacuna-100 for the CIFAR-10 / Lacuna-10 experiments, respectively [Golatkar et al., 2020a,b]. We run each experiment with 3 random seeds and report the mean and standard deviation.

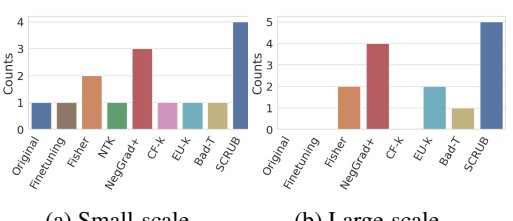

(a) Small-scale. (b) Large-scale.

Figure 2: The number of times each method was a top performer in forgetting in **RB-application**. A model is a top-performer if its 95% confidence interval overlaps with that of the best mean. Small-scale counts are over {ResNet, All-CNN} x {CIFAR, Lacuna} (4 total), large-scale additionally x {class, selective} (8 total). **SCRUB is the most consistent top-performer**.

We consider two settings throughout the paper: **small-scale** and **large-scale**. The former allows comparisons with NTK, which doesn't scale beyond it. For the small-scale, we exactly follow the setup in [Golatkar et al., 2020b] that uses only 5 classes from each of CIFAR and Lacuna ('CIFAR-5' / 'Lacuna-5'), with 100 train, 25 validation and 100 test examples per class. The forget set contains 25 examples from class 0 (5%). For the large-scale, we exactly follow the setup in [Golatkar et al., 2020a] that uses all 10 classes of each of CIFAR and Lacuna, and considers both a *class unlearning* scenario where the forget set is the entire training set for class 5 (10%), as well as a *selective unlearning* one where 100 examples of class 5 are forgotten (0.25% in CIFAR and 2% in Lacuna). We note that the Appendix contains complete results for all scenarios (several are ommitted from the main paper due to space constraints) and also contains the complete experimental details and hyperparameters.

**Unlearning algorithms** We compare against state-of-the-art approaches as well as various baselines: **Retrain**: Retraining from scratch without the forget set. This is assumed to not be viable in practice, but we include it as a reference. **Original**: The 'original' model trained on all of $\mathcal{D}$ without any unlearning. **Finetuning**: Finetunes the original model on data from the retain set $\mathcal{D}_r$. **NegGrad+**: Similar to Finetuning, starts from the original model and finetunes it both on data from the retain and forget sets, negating the gradient for the latter. Previous work considered a weaker baseline that only trains on the forget set, with a negative gradient (NegGrad). We tune the stronger NegGrad+ to achieve a good balance between the two objectives in the same way as SCRUB. **Fisher Forgetting** [Golatkar et al., 2020a]. **NTK Forgetting** [Golatkar et al., 2020b]. Catastrophic Forgetting-k (**CF-k**) and Exact Unlearning-k (**EU-k**) [Goel et al., 2022]: they freeze the first k layers of the original model and either finetune the remaining layers on $\mathcal{D}_r$ (CF-k) or train them from scratch on $\mathcal{D}_r$ (EU-k). **Bad-T**: the concurrent teacher-student method of [Chundawat et al., 2022]. We utilize the public code for the NTK and Fisher for correctness. We implement Bad-T ourselves (no code was available).

## 5.2 Unlearning for Removing Biases (RB)

In RB, we desire to achieve the highest possible error on $D_f$, without hurting the error of $D_r$ and $D_t$. Some example scenarios are removing biases from trained models, outdated or incorrect information. A maximal forget error on $D_f$ in this case is desirable as it reflects that we want the model to *never* make the predictions that were deemed harmful or no longer valid or accurate.

We report the results for this scenario in Figures 1 and 2. We also report complete results in the Appendix for all architectures, datasets and baselines. As shown in Figure 2, SCRUB is by far the most consistent method in terms of achieving high forget quality (highest forget error), while it doesn't hurt retain and test errors.

## 5.3 Unlearning for Resolving Confusion (RC)

In this section, following Goel et al. [2022], we study unlearning for resolving confusion between two classes that the original model suffers from due to a part of its training set being mislabelled. Specifically, we place all and only the mislabeled training examples in the forget set. A successful unlearning algorithm would thus entirely resolve the model's confusion. In more detail, the setup is: 1) Mislabel some portion of the training dataset (we mislabeled examples between classes 0 and 1 of each of CIFAR-5 and Lacuna-5), 2) Train the 'original model' on the (partly mislabeled) training dataset, and 3) Unlearn with the confused examples as the forget set.

We consider the following metrics inspired from Goel et al. [2022] that are variants of the model's error on examples from the confused classes (lower is better for both). We outline them below and define them formally in the Appendix.

- **Interclass Confusion IC-ERR** (e.g. IC test error, IC retain error). This counts mistakes where an example from either of the two confused classes was incorrectly predicted to belong to *any* other class.
- **FGT-ERR** (e.g. Fgt test error, Fgt retain error). This metric counts only misclassification *between the confused classes*, i.e. an example of class 0 being predicted as belonging to class 1, or vice versa.

We observe from Table 1 that, across the board, SCRUB is by far the most consistent method in terms of eliminating class confusion without damaging the quality of the model (on retain and test sets).

## 5.4 Unlearning for User Privacy (UP)

Next, we consider a privacy-critical application: the deletion of the data of a user exercising their right-to-be-forgotten. We use Membership Inference Attacks (MIAs) as the metric where the goal is that, after applying unlearning, an attacker can't tell apart examples that were unlearned from those that were truly never seen, thus protecting the privacy of the user requesting deletion.

While previous unlearning papers have reported MIA results [Golatkar et al., 2020a], those MIAs are simple and far from state-of-the-art MIAs used in privacy and security papers. Indeed, it's not trivial to adapt such MIAs to the unlearning protocol. To initiate bridging this gap, we report results on two MIAs: 1) a simple one that is closer to the typical MIAs used in the unlearning literature, and

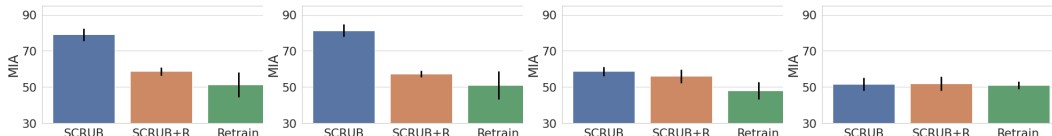

(a) Forget 100 examples. (b) Forget 150 examples. (c) Forget 175 examples. (d) Forget 200 examples.

Figure 3: MIA results for different sizes of forget set on CIFAR-10 with ResNet. Error bars show 95% confidence intervals. Rewinding is most useful for smaller forget sizes (see the discussion in Section 5.4). **SCRUB+R successfully defends MIAs, comparably to the Retrain oracle.**

2) the first, to our knowledge, adaptation of the state-of-the-art LiRA attack [Carlini et al., 2022] to the unlearning protocol. We describe both at a high-level here and in detail in the Appendix.

**Basic attack**  We train a binary classifier (the 'attacker') on the unlearned model's losses on forget and test examples for the objective of classifying 'in' (forget) versus 'out' (test). The attacker then makes predictions for held-out losses (losses the attacker wasn't trained on) that are balanced between forget and test losses. For a perfect defense, the attacker's accuracy is 50%, indicating that it is unable to separate the two sets, marking a success for the unlearning method.

We present Basic MIA results in Table 2 for CIFAR-10. We observe that SCRUB+R is a strong performer. EU-k can also perform strongly in this metric, though not as consistently. We also observe that rewinding was only triggered once, for *selective* unlearning. Recall that, in both selective and class unlearning, the forget set contains examples coming from a single class, but in the former, it doesn't contain *all* examples of that class; the rest are in the retain set. Thus, we indeed expect that rewinding would be more useful for selective unlearning: the smaller the portion of a class in the forget set, the larger the remainder of that class in the retain set and consequently, the better we expect the model to generalize on held-out examples of that class (lower error), making it in turn more likely to need to rewind, in order to also lower

|  | ResNet | | | | All-CNN | | | |
|---|---|---|---|---|---|---|---|---|
|  | Class | | Selective | | Class | | Selective | |
| Model | mean | std | mean | std | mean | std | mean | std |
| Retrain | 49.33 | 1.67 | 54.00 | 1.63 | 55.00 | 4.00 | 48.73 | 0.24 |
| Original | 71.10 | 0.67 | 65.33 | 0.47 | 66.50 | 0.50 | 71.40 | 0.70 |
| Finetune | 75.57 | 0.69 | 64.00 | 0.82 | 68.00 | 1.00 | 74.97 | 1.27 |
| NegGrad+ | 69.57 | 1.19 | 66.67 | 1.70 | 72.00 | 0.00 | 70.03 | 1.92 |
| CF-k | 75.73 | 0.34 | 65.00 | 0.00 | 69.00 | 2.00 | 72.93 | 1.06 |
| EU-k | **54.20** | 2.27 | **53.00** | 3.27 | 66.50 | 3.50 | **51.60** | 1.22 |
| Bad-T | **54.00** | 1.10 | 59.67 | 4.19 | 63.4 | 1.2 | 77.67 | 4.11 |
| SCRUB | **52.20** | 1.71 | 78.00 | 2.45 | **52.00** | 0.00 | **54.30** | 2.24 |
| SCRUB+R | **52.20** | 1.71 | 58.67 | 1.89 | **52.00** | 0.00 | **54.30** | 2.24 |

Table 2: **Basic MIA results (for UP): SCRUB successfully defends MIA**. Note the rewinding procedure of SCRUB+R was triggered only once; see discussion in Section 5.4 and Figure 3. In the Appendix, we also show that the forget error of SCRUB(+R) is close to that of Retrain, as desired.

the forget set error commensurately. In Figure 3 we plot results for different variants of selective unlearning, with different number of examples in the forget set (from the same class) and indeed find that for smaller numbers, rewinding is triggered. Our take-away is that, when needed, rewinding can substantially improve SCRUB's MIA results.

**LiRA-for-unlearning attack**  In the standard privacy setting, the LiRA attacker trains a large number of 'shadow models' [Carlini et al., 2022], for which it controls which examples are in the training set each time (by construction). To then predict the membership status of a 'target example', it estimates two Gaussian distributions: the distribution of confidences of that example under shadow models that trained on it, and the distribution of its confidences under shadow models that didn't. It predicts that the target example is 'in' if the likelihood of the former is larger than that of the latter.

We propose the first, to our knowledge, adaptation of LiRA for unlearning. This is a strong attack where we allow the attacker knowledge of the unlearning algorithm. Concretely, for each shadow model, the attacker also produces a 'shadow unlearned' model by applying the given unlearning algorithm several times, using a large number of forget sets (similar to Chen et al. [2021]). Now, for each 'target example', this setup allows the attacker to estimate a different pair of Gaussians: the distribution of (confidences of) that target example under models where it was *forgotten*, and as before, under models where it was not seen. The attacker predicts the example was forgotten if its likelihood under the former is larger than under the latter. We present all details and findings in the Appendix and observe that SCRUB+R outperforms the other methods in defending this attack.

## 5.5 Overview of results and take-aways

**Finetuning** is a poor method: it may retain model utility but fails to forget. The previous state-of-the-art **NTK** and **Fisher** models aren't among the top-performers either. The former performs poorly across the board in terms of forgetting and doesn't scale beyond small datasets. The latter sometimes performs well in RB in terms of forgetting, though not always and it is very slow (Figure 1), exceeding the runtime of Retrain; as also observed in [Goel et al., 2022]. **NegGrad+**, our proposed enhancement over the NegGrad baseline, is a strong baseline in terms of achieving a balance between forgetting and utility (it performs well in several settings in terms of RB, albeit not as consistently as SCRUB) and we encourage future work to report this baseline, though SCRUB outperforms it, especially for RC. **CF-k** inherits the performance profile of Finetune: it maintains utility but forgets poorly. On the other hand, **EU-k** more reliably forgets (performs strongly in UP, and often in RB), which is expected relative to CF-k due to retraining part of the network, but SCRUB outperforms it significantly in terms of RC and is more consistently a top performer in RB. A notable failure case that we discovered for EU-k in RB is *selective* unlearning (notice the contrast between EU-k's forget error between e.g. Figures 1a and 1b). This finding may speak to where class vs instance information is stored in neural networks and warrants further investigation. **Bad-T** is very efficient, but performs poorly in terms of forgetting, across all applications and also damages model utility. Overall, SCRUB is by far the most consistent in successfully forgetting under different metrics without hurting utility and being efficient.

In the Appendix, we discuss limitations of our work, broader impact, future work and present full experimental results and all details including hyperparameters and pseudocode.

## 6 Discussion and Conclusion

Despite substantial recent attention [Triantafillou et al., 2023], unlearning is a young area of research.

In this work, we propose SCRUB, an unlearning method that is unbound from limiting assumptions and poor scalability limits. SCRUB(+R) is found empirically to perform very strongly: it is by far the most consistent method in achieving good forgetting quality with respect to different application-dependent metrics, while incurring only minimal utility degradation. As such, we believe that SCRUB fills the important need for scalable unlearning methods that perform well in practice on several metrics of interest. However, important questions remain open for future work.

Crucially, despite progress in this direction, a well-established formal definition of the problem of unlearning remains elusive, and consequently we also lack well-established metrics for measuring (theoretically or empirically) the quality of unlearning algorithms. In this work, we focus on empirical evaluation with respect to different application-dependent metrics for forgetting quality that we believe are relevant in real-world applications. However, as the unlearning community matures, we hope that success criteria for (different applications of) unlearning will be formalized and standardized.

Further, an important limitation of SCRUB is the absence of theoretical guarantees, making it perhaps ill-suited for certain application scenarios. While methods that come with guarantees exist, they either aren't applicable to deep neural networks, or we find through our empirical analysis that they perform poorly and don't scale beyond small datasets. Therefore, future work that theoretically studies scalable methods like SCRUB would be very valuable. In absence of that, continuing to study and better understand the trade-offs offered by different methods is an important direction.

On the practical side, we hope that future work continues to push the limits of scalability and studies unlearning in larger models, different architectures, different training objectives like self-supervised learning and different domains and modalities, including foundation language models.

**Broader Impact**   While recent advances in deep learning represent exciting opportunities for our community, they also come with great responsibility. As researchers, we are responsible for understanding and mitigating the issues associated with the widespread use of deep learning technology. Machine learning models may carry harmful biases, unintended behaviours, or compromise user privacy. Our work is intended to take a step in addressing these issues via a post-processing 'unlearning' phase that makes progress over previous solutions in practice, as we show through an extensive empirical investigation. However, SCRUB does not come with theoretical guarantees and we can not prove that applying SCRUB perfectly mitigates those issues, so caution must be taken in practice and proper auditing of machine learning models is critical.

## Acknowledgments and Disclosure of Funding

We would like to thank Fabian Pedregosa for countless insightful discussions, especially on the topic of membership inference attacks.

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

# 7 Appendix

This Appendix contains the following sections:

- Section 8: Limitations and Future Work
- Section 9: Experimental Details and Pseudocode
- Section 10: Formal Description of Metrics
- Section 11: Membership Inference Attacks: Description and Additional Findings
- Section 12: Ablations and Sensitivity Analysis
- Section 13: Larger-scale settings
- Section 14: Additional Results for Removing Biases (RB)
- Section 15: Additional Results for Resolving Confusion (RC)
- Section 16: Additional Results for User Privacy (UP)

# 8 Limitations and Future Work

SCRUB has shown impressive results in terms of being consistently a top-performer in terms of unlearning, with a minimal drop in performance compared to previous works. However, SCRUB has limitations that we hope future work will address.

A significant step for future work is to develop theoretical guarantees for the gains provided by our methods. We opted to focus on an empirical approach for the following reasons: First, while theoretical guarantees abound for linear models, deep networks pose additional significant challenges. Second, methods accompanied with theoretical guarantees suffer from practical limitations with respect to accuracy and/or scalability. For these reasons we opted to approach the problem from a practical standpoint, pushing the envelope by developing unlearning algorithms which are top performers across many important different application scenarios, different evaluation metrics, different architectures and datasets. We look forward to future work that strives to strike a compromise between effective unlearning, good performance, scalability, and theoretical insights.

Another limitation of SCRUB is the difficulty and instability associated with tuning the min-max objective, as shown in the literature e.g. for GANs. For instance, this can lead to oscillating behaviour, as we show in Figure 6. We remedy this to a large extent in practice by providing a practical algorithm that works well, showing consistently improved results over prior work, but there is room for improvement on this front for future work.

SCRUB's rewinding procedure also has limitations. We find in practice throughout all of our experiments that it can help to substantially increase the success of SCRUB's defense on MIA in scenarios where the forget error obtained by SCRUB at the end of unlearning is 'too high'. However, a different failure case which can also appear is that SCRUB's forget error at the end of training is 'too low'. This can happen due to the way in which we tune hyperparameters, which is designed to not harm the retain and validation performance too much, and thus can in some cases lead to 'premature stopping' before the forget error reaches the same level as a reference point for how high it would be if the model had truly never seen those examples. We highlight that addressing all possible issues that can arise in all scenarios and provide an unlearning algorithm that performs strongly across the board is extremely challenging. The fact that we have observed failure cases for each algorithm, be it SCRUB or other baselines, is indicative of the extensiveness of the experimentation we conducted. Our work has made important strides in designing consistently strong-performing unlearning methods and we look forward to future contributions in this direction.

We hope that future work also continues to push the limits of scalability. We believe that our work constitutes an important step in this direction. However, the datasets and models we consider aren't too large, in order to allow comparisons to previous works that would not be feasible to run for larger scale experiments. An interesting topic of future work is investigating the interplay between SCRUB and other scalable algorithms like NegGrad+ with increasing amounts of scale.

Another really interesting future work direction is to investigate how different unlearning algorithms interact with different architectures, like Transformers, and loss functions, like self-supervised learning.

# 9 Experimental Details and Pseudocode

**Datasets**. We have used CIFAR-10 and Lacuna-10 datasets for evaluation purposes. CIFAR-10 consists of 10 classes with 60000 color images of size 32 x 32. In our experiments, the train, test, and validation sizes are 40000, 10000, and 10000 respectively. Lacuna-10 is a dataset derived from VGG-Faces [Cao and Yang, 2015]. We have followed the same procedure described in [Golatkar et al., 2020a] to build Lacuna. We randomly select 10 celebrities (classes) with at least 500 samples. We use 100 samples of each class to form the test-set, and the rest make the train-set. All the images are resized to 32 x 32. We also use CIFAR-100 and Lacuna-100 to pre-train the models. Lacuna-100 is built in a similar way as Lacuna-10, and there is no overlap between the two datasets. We have not applied any data augmentation throughout the experiments.

---

**Algorithm 1** SCRUB

---

**Require:** Teacher weights $w^o$
**Require:** Total max steps MAX-STEPS
**Require:** Total steps STEPS
**Require:** Learning rate $\epsilon$
  $w^u \leftarrow w^o$
  $i \leftarrow 0$
  **repeat**
    **if** $i <$ MAX-STEPS **then**
      $w^u \leftarrow$ DO-MAX-EPOCH$(w^u)$
    **end if**
    $w^u \leftarrow$ DO-MIN-EPOCH$(w^u)$
  **until** $i <$ STEPS

---

**Small-Scale datasets**. We followed the same procedure as decribed in [Golatkar et al., 2020b] to create the small versions of CIFAR-10 and Lacuna-10, namely CIFAR-5 and Lacuna-5. To this end, we take the first 5 classes of each dataset and randomly sample 100 images for each class. We make the train and test sets by sampling from the respective train and test sets of CIFAR-10 and Lacuna-10. We also make 25 samples from each class from the train set to create the validation sets.

**Models.** We use the same models with the same architectural modifications in [Golatkar et al., 2020a,b]. For All-CNN, the number of layers is reduced and batch normalization is added before each non-linearity. For Resnet, ResNet-18 architecture is used. For small scale experiments, the number of filters is reduced by 60% in each block. For the large-scale experiments, the exact architecture has been used.

---

**Algorithm 2** DO-MAX-EPOCH

---

**Require:** Student weights $w^u$
**Require:** Learning rate $\epsilon$
**Require:** Batch size B
**Require:** Forget set $D_f$
**Require:** Procedure NEXT-BATCH
  $b \leftarrow$ NEXT-BATCH$(D_f, \text{B})$
  **repeat**
    $w^u \leftarrow w^u + \epsilon \nabla_{w^u} \frac{1}{|b|} \sum_{x_f \in b} d(x_f; w^u)$
    $b \leftarrow$ NEXT-BATCH$(D_f, \text{B})$
  **until** $b$

---

**Pretraining.** Following the previous work for consistency, we apply pretraining. Specifically, for CIFAR datasets, we have pretrained the models on CIFAR-100. For Lacuna, we have pretrained the models on Lacuna-100. We pretrain the models for 30 epochs using SGD with a fixed learning rate of 0.1, Cross-Entropy loss function, weight decay 0.0005, momentum 0.9, and batch size 128.

**Baselines.** 'Original' is the model trained on the entire dataset $D$. For 'Retrain', we train the same architecture on $D_r$, with the same hyperparameters used during training of the original model. For 'Finetune', we fine-tune the 'original' model on $D_r$ for 10 epochs, with a fixed learning rate of 0.01 and weight-decay 0.0005. For 'NegGrad+', we fine-tune the 'original' model using the following loss:

$$\mathcal{L}(w) = \beta \times \frac{1}{|D_r|} \sum_{i=1}^{|D_r|} l(f(x_i; w), y_i) - (1 - \beta) \times \frac{1}{|D_f|} \sum_{j=1}^{|D_f|} l(f(x_j; w), y_j) \qquad (4)$$

Where $\beta \in [0, 1]$. We have tuned $\beta$ to get a high forget-error while not destroying retain-error and validation-error. For small-scale experiments, $\beta = 0.95$ and we have trained for 10 epochs, with SGD, 0.01 lr and 0.1 weight-decay. For large-scale experiments $\beta = 0.9999$ and we have trained for 5 epochs, with SGD, 0.01 lr, and 0.0005 weight-deay. Please note that small $\beta$ result in explosion quickly. For 'CF-k', we freeze the first k layers of the network and finetune the rest layers with $D_r$. We use the same setting as 'Finetune' baseline. For 'EU-k' we freeze the first k layers, and

| model | dataset | unlearning-type | forget-set bs | retain-set bs | max steps | min steps |
|---|---|---|---|---|---|---|
| ResNet | CIFAR-10 | class | 512 | 128 | 2 | 3 |
| | CIFAR-10 | selective | 16 | 64 | 5 | 5 |
| | Lacuna-10 | class | 128 | 128 | 5 | 5 |
| | Lacuna-10 | selective | 32 | 32 | 4 | 4 |
| | CIFAR-5 | selective | 32 | 32 | 10 | 10 |
| | Lacuna-5 | selective | 32 | 32 | 5 | 10 |
| All-CNN | CIFAR-10 | class | 512 | 256 | 3 | 4 |
| | CIFAR-10 | selective | 16 | 64 | 5 | 5 |
| | Lacuna-10 | class | 32 | 32 | 4 | 4 |
| | Lacuna-10 | selective | 8 | 32 | 2 | 4 |
| | CIFAR-5 | selective | 16 | 32 | 5 | 10 |
| | Lacuna-5 | selective | 32 | 32 | 5 | 10 |

Table 3: SCRUB's hyperparameters for each experiment

re-initialize the weights of the remaining layers and retrain them with $D_r$. As all the models are pretrained on larger datasets, for re-initializing we use the weights of the pretrained models. For 'EU-k' we use the same settings as the 'Retrain' baseline. In both 'EU-k' and 'CF-k' baselines, for both ResNet and All-CNN we freeze all the layers except for the last block of the network. For Resnet the last block is block4 and for All-CNN, the last block of layers is the 9th sequential block. For Bad-T, we follow the specifications given in Chundawat et al. [2022] with possible tuning of the parameters in different settings to get the highest forget-error without damaging retain-error. More specifically, for all models we perform one epoch of unlearning using Adam optimizer, and a temperature scalar of 4. Also, we use the whole retain-set compared to 30% reported in their paper as we empirically observed that using only 30% of retain-set for Bad-T yields high test errors.

---

**Algorithm 3** DO-MIN-EPOCH

---

**Require:** Student weights $w^u$
**Require:** Learning rate $\epsilon$
**Require:** Batch size B
**Require:** Retain set $D_r$
**Require:** Procedure NEXT-BATCH
  $b \leftarrow$ NEXT-BATCH$(D_r, \text{B})$
  **repeat**

  $$w^u \leftarrow w^u - \epsilon \nabla_{w^u} \frac{1}{|b|} \sum_{(x_r, y_r) \in b} \alpha d(x_r; w^u) +$$

  $\gamma l(f(x_r; w^u), y_r)$
  $b \leftarrow$ NEXT-BATCH$(D_r, \text{B})$
  **until** $b$

---

**SCRUB pseudocode and parameters.** We train SCRUB using Algorithm 1. Throughout the experiments, we tune the parameters to get a high forget-error while retaining the retain and validation error of the original model. We use the same optimizer for both min and max steps. We observed that for small-scale settings 'Adam' optimizer works better, while in large-scale settings both 'Adam' and 'SGD' could be used. For all experiments, we initialize the learning rate at 0.0005 and decay it by 0.1 after a number of min and max steps. Decaying the learning rate is crucial to control the oscillating behaviour of our min and max optimization. We apply a weight decay of 0.1 for small-scale setting and 0.0005 for large scale experiments, with a momentum of 0.9. Finally, we use different batch sizes for the forget-set and the retain-set to control the number of iteration in each direction, i.e the max and the min respectively. We report these in Table 3.

**System specification.** For scale-up experiments, the code is executed in Python 3.8, on an Ubuntu 20 machine with 40 CPU cores, a Nvidia GTX 2080 GPU and 256GB memory.

## 10 Formal Description of Metrics

In this section, we give more details and mathematical definitions of the metrics that we use throughout the paper. We first mathematically define the forget, retain and test errors, and then other application-dependent metrics, for Resolving Confusion (RC) and User Privacy (UP).

**Forget, retain and test errors** Here, we define the retain error, forget error and test error. Let $\mathcal{D}_r$, $\mathcal{D}_f$ and $\mathcal{D}_t$ denote the retain and forget portions of the training dataset, and a test dataset of heldout examples, respectively. We define error ($Err$) as follows:

$$Err(\mathcal{D}) = 1 - \frac{1}{|\mathcal{D}|} \sum_{(x_i, y_i) \in \mathcal{D}} \mathbb{1}[\arg\max(f(x_i; w)) == y_i] \qquad (5)$$

where $f$, parameterized by $w$ is the neural network model (comprised of a feature extractor followed by a softmax classifier layer), $\arg\max(f(x_i; w))$ is the label that the model thinks is most likely for example $x_i$, and $\mathbb{1}[x]$ is the indicator function that returns 1 if $x$ is True and 0 otherwise.

Based on the above, the retain error, forget error and test error are computed as $Err(\mathcal{D}_r)$, $Err(\mathcal{D}_f)$ and $Err(\mathcal{D}_t)$, respectively.

**Metrics for Unlearning for Resolving Confusion (RC)** We now define the class confusion metrics inspired by Goel et al. [2022]. Specifically, we explore a scenario where the forget set has confused labels (e.g. for two classes A and B, examples of A are labelled as B, and vice versa). The idea here is that, because mislabelled examples are only present in the forget set, successful unlearning (removing the influence of the forget set) would lead to a model that is not at all confused between classes A and B.

In more detail, the setup we follow is: 1) We first mislabel some portion of the training dataset (we mislabelled examples between classes 0 and 1 of each of CIFAR-5 and Lacuna-5 in our experiments), 2) train the 'original model' on the (partly mislabelled) training dataset (it has mislabelled examples for classes 0 and 1 but correct labels for the remaining classes), 3) perform unlearning where the forget set contains all and only the confused examples. Given this, the goal for the unlearning algorithm is to resolve the confusion of the original model.

We consider the following metrics (using terminology consistent with Goel et al. [2022]). They are presented in order of decreasing generality, and increasing focus on measuring degrees of confusion between the two classes considered.

- **Error** (e.g. test error, retain error, forget error). This counts all mistakes, so anytime that an example of some class is predicted to be in any other class, it will be counted. These are the same metrics that we use for the rest of the paper (see Equation 5). For test and retain error, lower is better, whereas for forget error, higher is better.
- **Interclass Confusion IC-ERR** (e.g. IC test error, IC retain error). This counts only mistakes that involve examples from the confused classes A and B. Specifically, it counts instances of any example of class A being predicted to be in *any* other class , and similarly for class B. Compared to Error, this metric is more focused towards understanding the result of the introduced confusion, since it only considers cases that relate to the confused classes. A successful unlearning method would make no such errors, so lower is better for each of IC test error and IC retain error.
- **FGT-ERR** (e.g. Fgt test error, Fgt retain error). This metric counts only misclassification *between the confused classes* A and B. Here, a mistake of an example of class A (or B) being predicted to be in class other than A or B will not be counted. Only mistakes of an example of class A being predicted to be in class B, and vice versa, are counted. **This is the most focused metric that directly measures the amount of remaining confusion between the two classes in question**. A successful unlearning method would make no such errors, so lower is better for each of Fgt test and Fgt retain.

More formally, Error is the same as defined in Equation 5. Let us now mathematically define IC-ERR and FGT-ERR. We denote by $C^{w,\mathcal{D}}$ the confusion matrix for model parameterized by $w$ on the dataset $\mathcal{D}$, and let $\mathcal{D}_A$ denote the part of the dataset $\mathcal{D}$ that belongs to class $A$. So, for example $\mathcal{D}_{r_A}$ denotes the part of the retain set $\mathcal{D}_r$ that belongs to class $A$, and the entry $C^{w,\mathcal{D}}_{A,B}$ of the confusion matrix stores the number of times that a sample belonging to class $A$ was (mis)classified as belonging to class $B$ in the dataset $\mathcal{D}$ by the model parameterized by $w$. Then, we have:

$$\text{IC-ERR}(\mathcal{D}, A, B; w) = \frac{\sum_k C^{w,\mathcal{D}}_{A,k} + \sum_{k'} C^{w,\mathcal{D}}_{B,k'}}{|\mathcal{D}_A| + |\mathcal{D}_B|} \qquad (6)$$

where $k \neq A, k' \neq B$.

So, for example, the 'IC test error' column in our tables is computed via IC-ERR$(\mathcal{D}_t, 0, 1; w)$, where $\mathcal{D}_t$ denotes the test set, and 0 and 1 are the two classes confused in our experiments. Analogously, 'IC retain error' is computed as IC-ERR$(\mathcal{D}_r, 0, 1; w)$

Finally:

$$\text{FGT-ERR}(\mathcal{D}, A, B; w) = C_{A,B}^{w,\mathcal{D}} + C_{B,A}^{w,\mathcal{D}} \tag{7}$$

That is, FGT-ERR only measures the misclassification between the two confused classes A and B. So, for example, the 'Fgt test error' in our tables is computed as FGT-ERR$(\mathcal{D}_t, 0, 1; w)$ and analogously 'Fgt retain error' is computed as FGT-ERR$(\mathcal{D}_r, 0, 1; w)$.

**User Privacy (UP) Metrics** Please see the next section for full details for each of the two Membership Inference Attacks (MIAs) that we use and experimental results.

## 11 Membership Inference Attacks: Description and Additional Findings

As mentioned in our paper, we utilize two different MIAs: 1) a 'Basic MIA' that is similar to the ones typically used in unlearning papers (but far from the state-of-the-art of MIAs used by privacy and security colleagues), and 2) the first, to our knowledge, adaptation of the state-of-the-art LiRA attack [Carlini et al., 2022] to the framework of unlearning ('LiRA-for-unlearning' MIA).

In this section, we use the term 'target model' to refer to the model that is being attacked and the term 'target example' to refer to an example whose membership status ('in' or 'out') the attacker tries to predict, based on the 'behaviour' (e.g loss value) of that example under the 'target model'. In both attacks that we consider, the target model is the unlearned model, and target examples are either forget set ('in') or test set ('out') examples. The unlearning algorithm successfully defends an MIA if the attacker can't tell apart examples that were unlearned (forget set examples) from examples that were truly never seen.

### 11.1 Basic MIA

Returning to our previous notation, let $l(f(x; w^u), y)$ denote the cross-entropy loss of the unlearned model (a deep network $f$ with weights $w^u$) on example $x$ with label $y$. We abbreviate this as $l(x, y)$ from now on; dropping the dependence on $f$ and $w^u$.

The attacker is a binary classifier that takes as input loss values, coming from either the forget set $\mathcal{D}_f$ or a held-out test set $\mathcal{D}_t$, and predicts whether the example whose loss value was presented was in the training set of the original model. We train this attacker via supervised learning on a class-balanced labelled training set for this binary problem: $\mathcal{D}_{train}^b = \{(l(x_i, y_i), y_i^b)\}$ where each $x_i$ is an example coming either from $D_f$ or $D_t$, and its binary label $y_i^b$ is defined as being 0, if $x_i \in \mathcal{D}_t$ and 1 if $x_i \in \mathcal{D}_f$. Once the binary classifier attacker is trained, we use it to make predictions for a held-out evaluation set of the binary problem: $\mathcal{D}_{eval}^b = \{(l(x_i, y_i), y_i^b)\}$ that is also balanced between examples coming from $\mathcal{D}_f$ and $\mathcal{D}_t$, but is disjoint from $\mathcal{D}_{train}^b$.

The attacker succeeds if it achieves high accuracy on $\mathcal{D}_{eval}^b$, meaning that it can tell apart examples that were part of the original training set from those that weren't, which marks a defeat for the unlearning model in terms of this metric, since it has 'left traces behind' (in this case, in terms of loss values) and leaks information about membership in the forget set. We consider that an optimal defense against this MIA corresponds to a 50% attack accuracy; that is, no better than randomly guessing whether an example had been trained on. In principle, the Retrain oracle should defend optimally: it in fact did not train on the forget set, so the forget and test sets are simply two different held-out sets for this model whose loss values should generally be indistinguishable from each other if these sets are identically-distributed. We find in our experiments, presented in the main paper, that SCRUB+R is able to defend this MIA comparably to Retrain, and outperforms the other baselines in its ability to consistently do so.

**Experimental details** In practice, if the distribution of the forget set and the test set are very different from each other, their loss values will be very distinguishable. This means that the binary classifier can tell them apart easily, but without having truly learned to infer membership in the

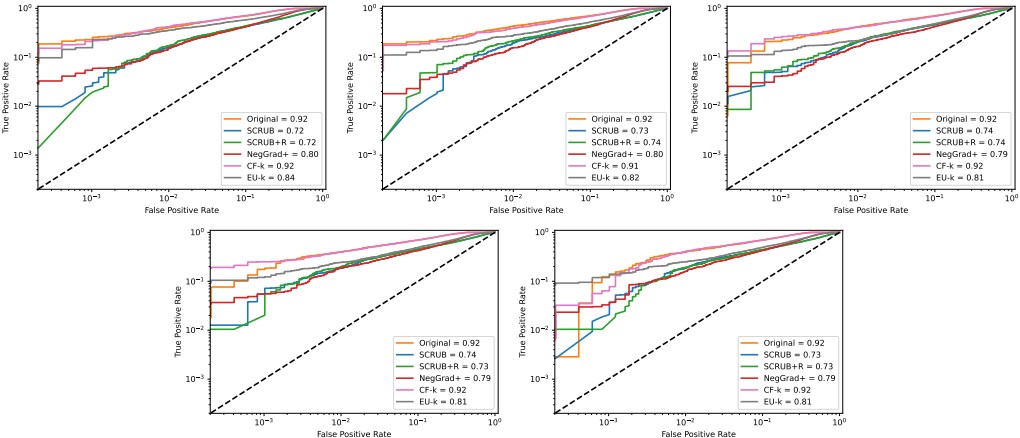

Figure 4: ROC curves for the strong LiRA-for-unlearning attack (Area Under the Curve (AUC) is also reported in the legend, for each unlearning method). Different subplots correspond to different target models (we trained the target unlearned model 5 times for each unlearning method, using different random seeds, and different forget sets). Positives are examples in the forget set, and negatives in the test set. A true positive means that the attacker correctly identified that an example was forgotten, whereas a false positive means that it incorrectly predicted that a test example was forgotten. We are primarily interested in the area of small False Positive Rate [Carlini et al., 2022] and a good unlearning method is associated with a smaller True Positive Rate, i.e. fewer successes for the attacker, especially in the region of interest. **We observe that SCRUB(+R) defends the strong LiRA-for-unlearning attack more successfully than the other baselines.**

training dataset. This makes the attacker's evaluation unreliable. To circumvent this problem, we ought to pick the held-out test set from the same distribution. More specifically, if the forget set is examples from the 'cat' class of CIFAR10 dataset, we use the same class for our held-out test set.

In our experiments, we clip the loss values to a range between [-400, +400] to remove anomalies. Also, we use the default LogisticRegression() classifier of the Python's scikit-learn library as our attack model, and perform a cross-validation with 5 random splits. We report the average accuracy of the evaluation part of each of the 5 folds as the MIA score. Ideally (for a perfect defense), this score is closest to 50%, indicating that the attacker fails to tell apart the forget set from the test set.

We present additional results for the Basic MIA attack in Section 16.

## 11.2 LiRA-for-unlearning attack

In the standard privacy setting, the LiRA attacker [Carlini et al., 2022] trains a large number of 'shadow models' [Shokri et al., 2017], for which it controls which examples are in the training set each time (by construction). To then predict the membership status of a 'target example', it estimates two Gaussian distributions, using the shadow models: the distribution of confidences of that example under shadow models that trained on it, and the distribution of its confidences under shadow models that didn't. Then, it computes the confidence of the target example under the target model and it predicts that the target example was 'in' if the likelihood of the target confidence under the former Gaussian is larger than that under the latter Gaussian.

Adapting LiRA to the framework of unlearning is not trivial, and we are not aware of this done in any previous work. **We propose the first, to our knowledge, adaptation of LiRA for unlearning**. This is a strong attack where we allow the attacker knowledge of the unlearning algorithm. Concretely, for each shadow model, the attacker also produces a 'shadow unlearned' model by applying the given unlearning algorithm several times, using a large number of forget sets (similar to Chen et al. [2021]). Now, for each 'target example', this setup allows the attacker to estimate a different pair of Gaussians: the distribution of (confidences of) that target example under models where it was *forgotten*, and as before, under models where it was not seen. The attacker then computes the confidence of the target example under the target model, and predicts the example was forgotten if its likelihood under the former is larger than under the latter.

**Experimental setup: overview** We run our LiRA-for-unlearning attack on selective unlearning on CIFAR-10, for the scenario where the forget set has size 200 and comes from class 5.

**Attacker:** For the attacker, we first train 256 'shadow original' models on random splits of half of the CIFAR-10 training set. Let $D$ denote the original dataset. To train each shadow model, we split $D$ in half, and use one half to as the 'training set' and the other half as the 'test set' of that particular shadow model. Then, for each of these 'shadow original' models, we run unlearning on 10K different forget sets (for each unlearning method). Specifically, the forget set is a random subset of 200 examples of class 5, sampled from the training set of the corresponding 'shadow original' model. After this procedure, for every example in class 5, we select 256 associated shadow models when it was not included in training, and 256 shadow models when it was unlearned (i.e. it was in the forget set). After this, the LiRA attack proceeds as normal, where we take each in / out distribution and apply a likelihood ratio test on an unknown example to infer membership.

**Defender:** We next train the target model that LiRA-for-unlearning will attack. For this, we begin by training the 'original' model, on (a

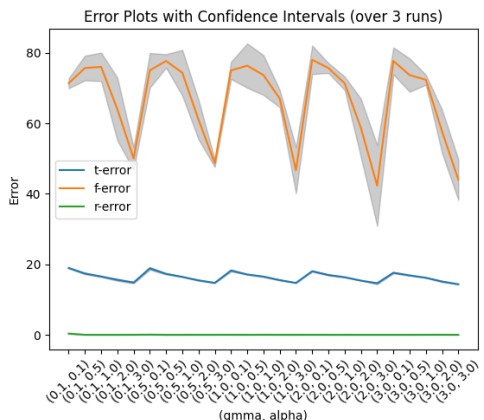

Figure 5: Sensitivity of SCRUB to $\gamma$ and $\alpha$. To create this plot, we ran SCRUB many times for different values of $\gamma$ ([0.1, 0.5, 1, 2, 3]) and $\alpha$ ([0.1, 0.5, 1, 2, 3]). The x-axis is represents combinations of these values. t-error, f-error and r-error refer to test, forget and retain error, respectively. We find that SCRUB is not very sensitive to these hyperparameters: the retain error remains low across values, and there are several different settings to these hyperparameters for which we can obtain the desired results for test and forget errors too.

random split of) half of $D$. Then, we apply the given unlearning algorithm on a randomly-sampled forget set, which is a subset of the original model's training set of size 200, coming from class 5, in the same way as was done by the shadow models.

**Implementation note** We run this attack at a much larger scale than the remaining experiments of the paper (we run 10K unlearning runs, on different forget sets, for *each* of the 256 'shadow original' models). We do this because the strength of the attacker is heavily dependent on the number of shadow original/unlearned models, and we wanted to benchmark our baselines against a very strong attacker. Therefore, to allow better scaling, instead of implementing SCRUB+R as rewinding, we implement it through filtering runs of SCRUB that don't satisfy the condition of SCRUB+R that the forget set error should be close to the validation error (where, as explained in the main paper, this refers to the validation set that is constructed to have the same distribution as the forget set; containing examples of only the same class as the one in the forget set). We used 0.1 as our threshold.

**Computing the 'confidence'** Consistent with [Carlini et al., 2022], the confidence of an example $(x, y)$ (where $y$ denotes the ground-truth class label) is defined as $\mathrm{softmax}(f(x))[y]$. In words, the confidence is the softmax probability of the *correct class*. Following Carlini et al. [2022], we apply logit-scaling to each confidence, to make their distributions Gaussian.

**Conclusions and findings** Figure 4 plots the ROC curve, showing the False Positive Rate and True Positive Rate of the attacker, in log-log scale. Different subplots correspond to different target unlearned models, each of which was trained with a different random seed, and different retain/forget set split. A successful defense is associated with a smaller Area Under the Curve (AUC); meaning fewer True Positives for the attacker. Carlini et al. [2022] however advocate that the AUC is not a good indicator of the attacker's strength and, instead, they argue that we should primarily consider the region of the ROC curve associated with very small False Positive Rates. We observe that, especially in that region, SCRUB(+R) is the strongest method in terms of defending our LiRA-for-unlearning attack (and also we observe that SCRUB(+R) has the best AUC too). The improved NegGrad+ baseline that we also proposed in this paper is also a strong model in terms of defending this attack.

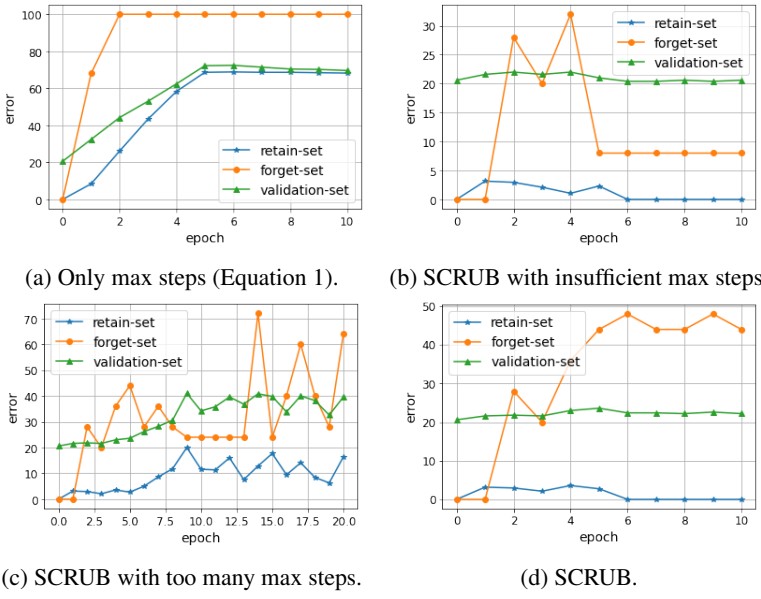

(a) Only max steps (Equation 1).    (b) SCRUB with insufficient max steps.

(c) SCRUB with too many max steps.    (d) SCRUB.

Figure 6: Illustration of training dynamics of SCRUB variants, on CIFAR-5 with a ResNet model. Performing the right interleaving of *min-steps* and *max-steps* is important for achieving a good balance between high forget error and low retain and validation errors.

We found that CF-k is not able to improve the privacy of the original model in most cases, while EU-k can sometimes improve but only slightly, and not reliably.

**Limitations**    To stay consistent with previous work on unlearning, as mentioned previously, we turn off data augmentations. Consequently, the 'original model' (before unlearning is applied) has overfitted more than a state-of-the-art CIFAR model would. Indeed, as can be seen from Figure 4, the 'Original' model has poor privacy (the attacker has a high True Positive Rate). We note that this is the first, to our knowledge, investigation of privacy of unlearning algorithms using strong MIAs, and we hope that future work continues to investigate increasingly more realistic scenarios with models closer to the state-of-the-art, and considers unlearning on original models of varying degrees of privacy and generalization ability.

## 12    Ablations and Sensitivity Analysis

In this section, we illustrate the training dynamics of SCRUB and the importance of different design choices. As a reminder, the student is initialized from the teacher and subsequently undergoes an alternating sequence of *max-steps* and *min-steps*; the former encouraging the student to move far from the teacher on the forget set, and the latter encouraging it to stay close to the teacher on the retain set. We also found it useful to perform a sequence of additional *min-steps* after the alternating sequence. We now explore the effect of these decisions.

First, we show that performing only *max-steps*, by optimizing Equation 1, is not a good solution. Simply pushing the student away from the teacher on the forget set achieves forgetting but unfortunately also hurts the retain and validation set performance (Figure 6a). Therefore, alternating between *max-steps* and *min-steps* is necessary. However, it is important to find the right balance. For instance, as seen in Figure 6b, performing too few *max-steps* leads to the unwanted consequence of the forget error dropping. On the other hand, removing the final sequence of only *min-steps* is also harmful, as shown in Figure 6c that trains for a larger number of epochs of an equal number of (alternating) *max-steps* and *min-steps* without achieving a good balance at any point throughout the trajectory. On the other hand, SCRUB (Figure 6d) achieves a good balance of high forget error and low retain and validation error simultaneously. We also ablate the cross-entropy term in Equation 3, which provides a small but consistent added protection against degrading performance in Figure 10. We show additional examples of training dynamics (Figures 7, 8, 9).

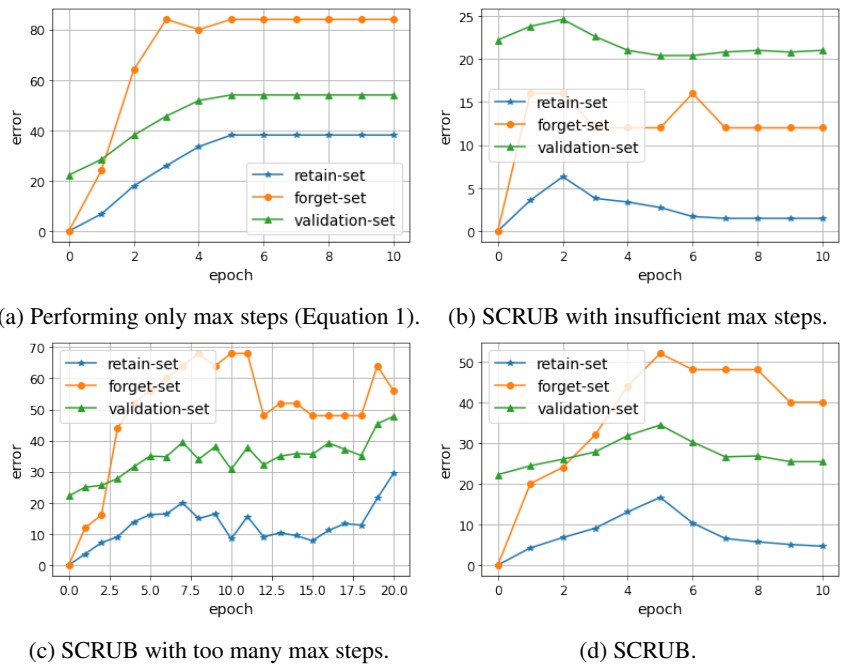

(a) Performing only max steps (Equation 1).     (b) SCRUB with insufficient max steps.

(c) SCRUB with too many max steps.         (d) SCRUB.

Figure 7: Illustration of training dynamics of SCRUB variants, on CIFAR-5 with All-CNN. Performing the right interleaving of *min-steps* and *max-steps* is important for achieving a good balance between high forget error and low retain and validation errors.

Table 4: Results for class-unlearning. Class-0 (400 examples) of Cifar-100 (40k examples). The numbers are averaged over 3 runs.

| Model | Test error ($\downarrow$) | | Retain error ($\downarrow$) | | Forget error ($\uparrow$) | | Basic MIA | |
|---|---|---|---|---|---|---|---|---|
| | mean | std | mean | std | mean | std | mean | std |
| Original | 35.84 | 0.76 | 0.07 | 0.02 | 0.00 | 0.00 | 61.67 | 16.20 |
| Retrain | 36.92 | 1.02 | 0.09 | 0.06 | 100.00 | 0.00 | 55.67 | 34.09 |
| NegGrad+ | 43.16 | 3.13 | 6.86 | 5.16 | 49.50 | 12.79 | 64.67 | 18.07 |
| Bad-T@epoch1 | 44.14 | 1.84 | 8.78 | 2.08 | 63.83 | 24.64 | 61.67 | 20.57 |
| Bad-T@epoch10 | 43.33 | 1.27 | 7.74 | 3.21 | 85.33 | 4.18 | 63.33 | 28.15 |
| SCRUB+R | 37.60 | 1.18 | 0.12 | 0.10 | 100.00 | 0.00 | 57.33 | 28.99 |
| SCRUB | 38.27 | 3.08 | 0.78 | 2.47 | 100.00 | 0.00 | 48.33 | 12.25 |

Finally, we also investigate the sensitivity of SCRUB's results on the $\gamma$ and $\alpha$ hyperparameters, in Figure 5. We find that SCRUB is not very sensitive to these hyperparameters: the retain error remains low across values, and there are several different settings to these hyperparameters for which we can obtain the desired results for test and forget errors too.

## 13   Larger-scale Settings

In this section, we scale up to an even larger setting, by choosing a larger model and performing a larger classification task. The experiments here are conducted on VGG16+BN with almost 138M parameters, and CIFAR-100, with varying forget set sizes. The results are reported in Table 4 and Table 5. Furthermore, for these experiments we train Bad-T for different number of iterations (recall that Bad-T originally proposes 1 epoch of unlearning) to study if its performance would improve with longer iterations. The results suggest that, again, SCRUB is consistently among the top performers across all the metrics. Furthermore, the results show that longer iterations does not improve Bad-T in any of the metrics.

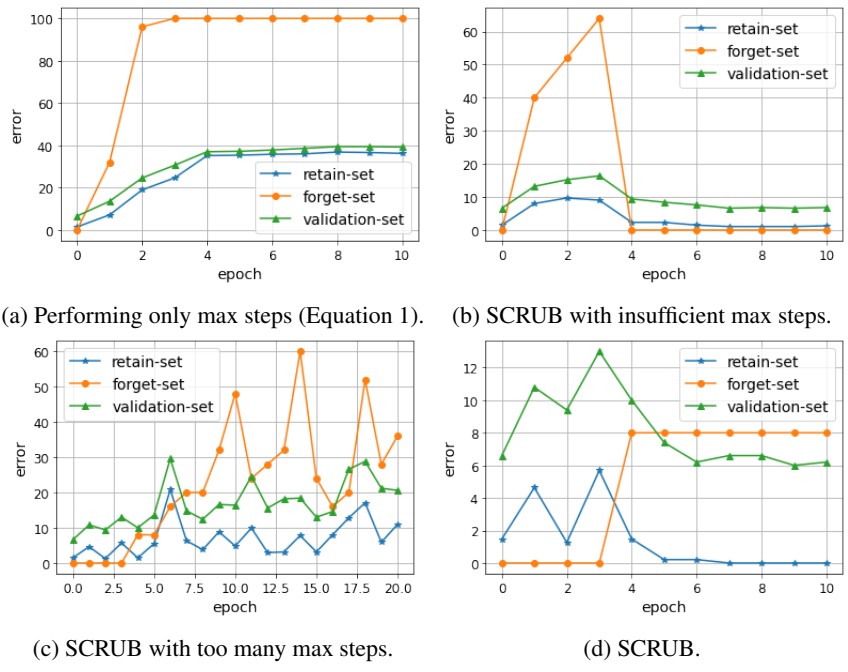

(a) Performing only max steps (Equation 1).    (b) SCRUB with insufficient max steps.

(c) SCRUB with too many max steps.    (d) SCRUB.

Figure 8: Illustration of training dynamics of SCRUB variants, on Lacuna-5 with ResNet. Performing the right interleaving of *min-steps* and *max-steps* is important for achieving a good balance between high forget error and low retain and validation errors.

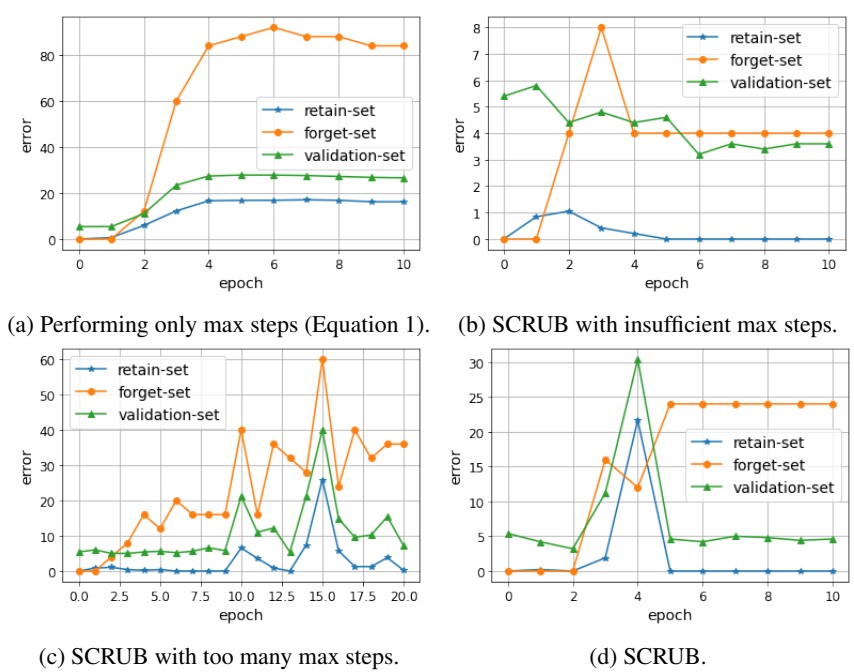

(a) Performing only max steps (Equation 1).    (b) SCRUB with insufficient max steps.

(c) SCRUB with too many max steps.    (d) SCRUB.

Figure 9: Illustration of training dynamics of SCRUB variants, on Lacuna-5 with All-CNN. Performing the right interleaving of *min-steps* and *max-steps* is important for achieving a good balance between high forget error and low retain and validation errors.

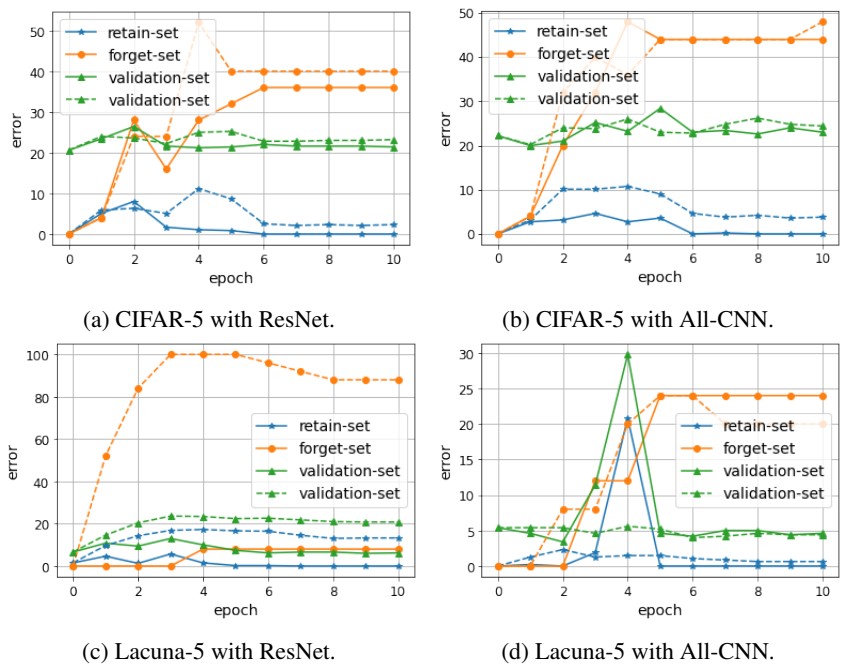

(a) CIFAR-5 with ResNet.

(b) CIFAR-5 with All-CNN.

(c) Lacuna-5 with ResNet.

(d) Lacuna-5 with All-CNN.

Figure 10: Effect of adding the cross-entropy loss in Equation 3. Dashed lines omit cross-entropy while solid lines use it. We find that the addition of cross-entropy offers an additional protection to maintaining the model's performance during the unlearning procedure. This sometimes comes at the cost of smaller forget set error, compared to the forget set error that would have been achieved if cross-entropy was omitted from the loss.

Table 5: Results for selective unlearning. 10 examples from class-0 of Cifar-100 dataset. The numbers are averaged over 10 runs.

| Forget Size | Model | Test error (↓) mean | std | Retain error (↓) mean | std | Forget error (↑) mean | std | Basic MIA mean | std |
|---|---|---|---|---|---|---|---|---|---|
| | Original | 37.46 | 4.99 | 0.54 | 1.74 | 0.00 | 0.00 | 58.00 | 9.10 |
| | Retrain | 37.54 | 5.24 | 0.57 | 1.84 | 19.00 | 15.81 | 45.00 | 19.04 |
| | NegGrad+ | 45.98 | 6.79 | 11.64 | 12.02 | 44.00 | 11.15 | 55.00 | 23.88 |
| 10 | Bad-T@epoch1 | 47.23 | 5.76 | 13.79 | 10.28 | 66.00 | 18.22 | 62.00 | 30.20 |
| | Bad-T@epoch10 | 47.09 | 5.65 | 13.80 | 9.79 | 96.00 | 8.07 | 77.00 | 17.25 |
| | SCRUB+R | 38.40 | 4.70 | 0.69 | 2.07 | 40.00 | 13.33 | 61.00 | 17.59 |
| | SCRUB | 38.18 | 4.78 | 0.48 | 1.44 | 45.00 | 13.60 | 56.00 | 19.76 |
| | Original | 37.46 | 4.99 | 0.54 | 1.74 | 0.00 | 0.00 | 63.00 | 10.96 |
| | Retrain | 37.74 | 5.31 | 0.56 | 1.77 | 19.50 | 12.00 | 49.00 | 11.47 |
| | NegGrad+ | 45.56 | 7.00 | 11.14 | 12.57 | 47.00 | 9.89 | 67.50 | 13.94 |
| 20 | Bad-T@epoch1 | 46.46 | 6.28 | 12.95 | 11.71 | 55.50 | 16.20 | 55.00 | 21.59 |
| | Bad-T@epoch10 | 45.88 | 5.70 | 12.03 | 9.44 | 92.00 | 10.25 | 70.00 | 10.87 |
| | SCRUB+R | 38.39 | 4.63 | 0.68 | 2.00 | 23.00 | 8.70 | 49.50 | 11.68 |
| | SCRUB | 38.29 | 4.61 | 0.49 | 1.42 | 29.00 | 16.28 | 54.50 | 18.54 |

| Model | CIFAR-10 | | | | Lacuna-10 | | | |
|---|---|---|---|---|---|---|---|---|
| | ResNet | | All-CNN | | ResNet | | All-CNN | |
| | class | selective | class | selective | class | selective | class | selective |
| Finetune | 3.8 | 3.09 | 3.33 | 3.03 | 1.7 | 2.03 | 2.16 | 2.00 |
| Fisher | 0.08 | 0.07 | 0.16 | 0.14 | 0.08 | 0.07 | 0.16 | 0.15 |
| NegGrad+ | 3.4 | 2.96 | 2.30 | 2.97 | 1.66 | 1.5 | 2.41 | 2.27 |
| CF-k | 3.55 | 3.17 | 3.37 | 2.91 | 3.42 | 3.20 | 3.27 | 3.11 |
| EU-k | 1.41 | 1.26 | 1.34 | 1.20 | 1.39 | 1.28 | 1.32 | 1.26 |
| Bad-T | 19.07 | 20.44 | 17.91 | 17.03 | 20.05 | 20.27. | 16.32 | 16.02 |
| SCRUB | 7.84 | 7.41 | 6.36 | 5.33 | 2.17 | 1.95 | 2.81 | 2.48 |

Table 6: **Scale-up factor**: the fraction of the runtime of retrain from scratch over the runtime of each given unlearning algorithm. That is, a scale-up value of X for an unlearning algorithm means that that algorithm runs X times faster than retrain from scratch.

| Model | CIFAR-5 | | | | | | Lacuna-5 | | | | | |
|---|---|---|---|---|---|---|---|---|---|---|---|---|
| | Test error ($\downarrow$) | | Retain error ($\downarrow$) | | Forget error ($\uparrow$) | | Test error ($\downarrow$) | | Retain error ($\downarrow$) | | Forget error ($\uparrow$) | |
| | mean | std | mean | std | mean | std | mean | std | mean | std | mean | std |
| Retrain | 24.9 | 2.5 | 0.0 | 0.0 | 28.8 | 5.9 | 5.8 | 0.4 | 0.0 | 0.0 | 4.8 | 3.4 |
| Original | 24.2 | 2.6 | 0.0 | 0.0 | 0.0 | 0.0 | 5.7 | 0.4 | 0.0 | 0.0 | 0.0 | 0.0 |
| Finetune | 24.3 | 2.4 | 0.0 | 0.0 | 0.0 | 0.0 | 5.6 | 0.3 | 0.0 | 0.0 | 0.0 | 0.0 |
| Fisher | 31.6 | 3.4 | 14.0 | 6.0 | 4.8 | 5.2 | 14.0 | 3.6 | 6.7 | 3.3 | 6.4 | 8.3 |
| NTK | 24.4 | 2.6 | 0.0 | 0.0 | 22.4 | 9.2 | 5.6 | 0.5 | 0.0 | 0.0 | 0.0 | 0.0 |
| NegGrad+ | 25.5 | 1.1 | 0.0 | 0.0 | 41.3 | 6.1 | 6.1 | 0.7 | 0.0 | 0.0 | 1.3 | 2.3 |
| CF-k | 22.6 | 1.9 | 0.0 | 0.0 | 0.0 | 0.0 | 5.8 | 0.4 | 0.0 | 0.0 | 0.0 | 0.0 |
| EU-k | 23.5 | 1.1 | 0.0 | 0.0 | 10.7 | 2.3 | 5.9 | 0.6 | 0.0 | 0.0 | 0.0 | 0.0 |
| Bad-T | 27.73 | 1.89 | 5.12 | 1.56 | 8.00 | 8.64 | 5.00 | 0.33 | 0.14 | 0.10 | 0.00 | 0.00 |
| SCRUB | 24.2 | 1.6 | 0.0 | 0.0 | 40.8 | 1.8 | 6.2 | 0.73 | 0.0 | 0.0 | 24.8 | 5.2 |

Table 7: **Small-scale** results with ResNet for the **Removing Biases (RB)** application. SCRUB is the top-performer in terms of forgetting with minimal performance degradation.

# 14 Additional Results for Removing Biases (RB)

In this section, we provide the results for all scenarios we studied for the **Removing Biases (RB)** application for ResNet and All-CNN, on both CIFAR and Lacuna, for both small-scale and large-scale, for completeness, in Tables 7, 8, 9, 10, 11, 12.

| Model | CIFAR-5 | | | | | | Lacuna-5 | | | | | |
|---|---|---|---|---|---|---|---|---|---|---|---|---|
| | Test error ($\downarrow$) | | Retain error ($\downarrow$) | | Forget error ($\uparrow$) | | Test error ($\downarrow$) | | Retain error ($\downarrow$) | | Forget error ($\uparrow$) | |
| | mean | std | mean | std | mean | std | mean | std | mean | std | mean | std |
| Retrain | 24.36 | 1.61 | 0.13 | 0.28 | 28.8 | 9.12 | 4.6 | 0.38 | 0.0 | 0.0 | 4.67 | 6.41 |
| Original | 24.08 | 1.86 | 0.17 | 0.38 | 0.0 | 0.0 | 4.53 | 0.47 | 0.0 | 0.0 | 0.0 | 0.0 |
| Finetune | 23.48 | 1.91 | 0.04 | 0.09 | 0.0 | 0.0 | 9.77 | 10.76 | 6.63 | 13.22 | 19.33 | 40.03 |
| Fisher | 42.64 | 6.56 | 31.83 | 10.47 | 15.2 | 16.83 | 52.53 | 13.87 | 51.09 | 14.54 | 39.33 | 40.43 |
| NTK | 24.16 | 1.77 | 0.17 | 0.38 | 13.6 | 8.29 | 4.47 | 0.47 | 0.0 | 0.0 | 3.33 | 4.68 |
| NegGrad+ | 26.07 | 1.21 | 0.56 | 0.49 | 36.00 | 10.58 | 5.27 | 0.76 | 0.14 | 0.12 | 12.00 | 13.86 |
| CF-k | 22.67 | 1.55 | 0.00 | 0.00 | 0.00 | 0.00 | 4.67 | 0.70 | 0.00 | 0.00 | 0.00 | 0.00 |
| EU-k | 25.87 | 0.64 | 3.23 | 1.69 | 8.00 | 6.93 | 5.20 | 0.20 | 0.00 | 0.00 | 0.00 | 0.00 |
| Bad-T | 25.87 | 1.80 | 9.68 | 0.45 | 10.67 | 4.99 | 8.87 | 0.66 | 2.32 | 0.79 | 0.00 | 0.00 |
| SCRUB | 23.88 | 1.78 | 0.08 | 0.12 | 40.8 | 8.2 | 3.87 | 0.72 | 0.0 | 0.0 | 25.33 | 4.13 |

Table 8: **Small-scale** results with All-CNN for the **Removing Biases (RB)** application. SCRUB is the top-performer in terms of forgetting with minimal performance degradation.

| Model | CIFAR-10 | | | | | | Lacuna-10 | | | | | |
|---|---|---|---|---|---|---|---|---|---|---|---|---|
| | Test error (↓) | | Retain error (↓) | | Forget error (↑) | | Test error (↓) | | Retain error (↓) | | Forget error (↑) | |
| | mean | std | mean | std | mean | std | mean | std | mean | std | mean | std |
| Retrain | 14.72 | 0.16 | 0.0 | 0.0 | 100.0 | 0.0 | 2.87 | 0.34 | 0.0 | 0.0 | 99.75 | 0.56 |
| Original | 16.56 | 0.1 | 0.0 | 0.0 | 0.0 | 0.0 | 3.07 | 0.26 | 0.0 | 0.0 | 0.0 | 0.0 |
| Finetune | 16.41 | 0.09 | 0.0 | 0.0 | 0.0 | 0.0 | 3.02 | 0.37 | 0.0 | 0.0 | 0.0 | 0.0 |
| Fisher | 26.42 | 1.41 | 2.45 | 0.84 | 100.0 | 0.0 | 3.33 | 0.54 | 0.0 | 0.0 | 100.0 | 0.0 |
| NegGrad+ | 17.84 | 1.46 | 1.74 | 2.55 | 91.26 | 7.73 | 3.41 | 0.17 | 0.00 | 0.00 | 14.90 | 1.78 |
| CF-k | 15.31 | 0.12 | 0.00 | 0.00 | 0.03 | 0.01 | 2.89 | 0.22 | 0.00 | 0.00 | 0.00 | 0.00 |
| EU-k | 18.73 | 0.42 | 0.00 | 0.00 | 98.79 | 0.18 | 3.19 | 0.17 | 0.01 | 0.02 | 4.06 | 0.83 |
| Bad-T | 19.56 | 1.44 | 11.34 | 1.82 | 94.67 | 6.12 | 3.37 | 0.50 | 1.06 | 0.47 | 67.60 | 24.26 |
| SCRUB | 15.73 | 0.17 | 0.51 | 0.02 | 100.0 | 0.0 | 3.69 | 0.36 | 0.28 | 0.23 | 100.0 | 0.0 |

Table 9: **Large-scale, class unlearning** results with ResNet for the **Removing Biases (RB)** application. SCRUB and EU-k are the top-performers in this setting in terms of forgetting with minimal performance degradation. Note, however, that EU-k doesn't perform strongly across the board and in particular performs very poorly in selective unlearning (notice the contrast between EU-k's forget error between Figures 1a and 1b Fisher is also a top-performer in terms of forget error in this setting too, but on CIFAR causes a large degradation in test error, as is often observed for this method.

| Model | CIFAR-10 | | | | | | Lacuna-10 | | | | | |
|---|---|---|---|---|---|---|---|---|---|---|---|---|
| | Test error (↓) | | Retain error (↓) | | Forget error (↑) | | Test error (↓) | | Retain error (↓) | | Forget error (↑) | |
| | mean | std | mean | std | mean | std | mean | std | mean | std | mean | std |
| Retrain | 13.97 | 0.19 | 0.0 | 0.0 | 100.0 | 0.0 | 1.59 | 0.36 | 0.0 | 0.0 | 100.0 | 0.0 |
| Original | 15.56 | 0.25 | 0.0 | 0.0 | 0.0 | 0.0 | 1.56 | 0.33 | 0.0 | 0.0 | 0.0 | 0.0 |
| Finetune | 15.39 | 0.22 | 0.0 | 0.0 | 0.0 | 0.0 | 1.67 | 0.44 | 0.0 | 0.0 | 0.0 | 0.0 |
| Fisher | 27.4 | 2.28 | 3.66 | 1.03 | 99.0 | 0.0 | 1.78 | 0.29 | 0.0 | 0.0 | 89.0 | 0.0 |
| NegGrad+ | 17.87 | 0.31 | 0.58 | 0.13 | 87.22 | 1.67 | 1.63 | 0.17 | 0.00 | 0.00 | 6.56 | 1.13 |
| CF-k | 14.99 | 0.23 | 0.00 | 0.00 | 0.00 | 0.00 | 1.48 | 0.36 | 0.00 | 0.00 | 0.00 | 0.00 |
| EU-k | 15.30 | 0.69 | 0.13 | 0.14 | 100.00 | 0.00 | 1.74 | 0.45 | 0.00 | 0.00 | 77.19 | 39.51 |
| Bad-T | 16.98 | 0.40 | 5.84 | 0.43 | 81.93 | 3.50 | 2.56 | 0.09 | 0.37 | 0.18 | 38.65 | 36.80 |
| SCRUB | 15.06 | 0.14 | 0.12 | 0.03 | 100.0 | 0.0 | 2.0 | 0.4 | 0.0 | 0.0 | 100.0 | 0.0 |

Table 10: **Large-scale, class unlearning** results with All-CNN for the **Removing Biases (RB)** application. SCRUB is the top-performer in terms of forgetting with minimal performance degradation.

| Model | CIFAR-10 | | | | | | Lacuna-10 | | | | | |
|---|---|---|---|---|---|---|---|---|---|---|---|---|
| | Test error (↓) | | Retain error (↓) | | Forget error (↑) | | Test error (↓) | | Retain error (↓) | | Forget error (↑) | |
| | mean | std | mean | std | mean | std | mean | std | mean | std | mean | std |
| Retrain | 17.4 | 0.14 | 0.0 | 0.0 | 29.67 | 3.21 | 2.7 | 0.2 | 0.0 | 0.0 | 1.0 | 1.0 |
| Original | 17.36 | 0.14 | 0.0 | 0.0 | 0.0 | 0.0 | 2.73 | 0.15 | 0.0 | 0.0 | 0.0 | 0.0 |
| Finetune | 17.37 | 0.11 | 0.0 | 0.0 | 0.0 | 0.0 | 2.63 | 0.12 | 0.0 | 0.0 | 0.0 | 0.0 |
| Fisher | 21.23 | 0.27 | 2.88 | 0.54 | 3.0 | 2.65 | 3.1 | 0.35 | 0.0 | 0.0 | 0.0 | 0.0 |
| NegGrad+ | 22.7 | 0.6 | 4.1 | 0.5 | 53.7 | 6.8 | 4.7 | 0.2 | 0.9 | 0.1 | 13.0 | 1.0 |
| CF-k | 17.4 | 0.1 | 0.0 | 0.0 | 0.0 | 0.0 | 2.7 | 0.2 | 0.0 | 0.0 | 0.0 | 0.0 |
| EU-k | 21.8 | 0.2 | 0.4 | 0.6 | 23.7 | 3.5 | 2.9 | 0.1 | 0.0 | 0.0 | 0.0 | 0.0 |
| Bad-T | 23.47 | 1.57 | 14.53 | 1.65 | 34.67 | 1.70 | 7.30 | 2.20 | 3.26 | 1.83 | 0.33 | 0.47 |
| SCRUB | 18.04 | 0.2 | 0.0 | 0.0 | 70.33 | 4.16 | 3.0 | 0.0 | 0.0 | 0.0 | 4.67 | 3.06 |

Table 11: **Large-scale, selective unlearning** results with ResNet for the **Removing Biases (RB)** application. SCRUB and NegGrad+ are the top-performers in terms of forgetting, though NegGrad+ has worse test performance than SCRUB in both cases. Note also that NegGrad+ isn't as consistent at forgetting across settings as SCRUB.

| Model | CIFAR-10 | | | | | | Lacuna-10 | | | | | |
|---|---|---|---|---|---|---|---|---|---|---|---|---|
| | Test error (↓) | | Retain error (↓) | | Forget error (↑) | | Test error (↓) | | Retain error (↓) | | Forget error (↑) | |
| | mean | std | mean | std | mean | std | mean | std | mean | std | mean | std |
| Retrain | 16.47 | 0.21 | 0.0 | 0.0 | 25.67 | 2.31 | 1.6 | 0.44 | 0.0 | 0.0 | 0.67 | 0.58 |
| Original | 16.43 | 0.08 | 0.0 | 0.0 | 0.0 | 0.0 | 1.53 | 0.31 | 0.0 | 0.0 | 0.0 | 0.0 |
| Finetune | 16.5 | 0.18 | 0.0 | 0.0 | 0.0 | 0.0 | 1.43 | 0.21 | 0.0 | 0.0 | 0.0 | 0.0 |
| Fisher | 21.39 | 1.22 | 4.0 | 1.44 | 13.0 | 11.27 | 1.87 | 0.21 | 0.01 | 0.02 | 0.0 | 0.0 |
| NegGrad+ | 21.36 | 0.34 | 3.23 | 0.37 | 45.33 | 2.89 | 2.77 | 0.25 | 0.40 | 0.07 | 8.67 | 0.58 |
| CF-k | 16.29 | 0.07 | 0.00 | 0.00 | 0.00 | 0.00 | 1.53 | 0.31 | 0.00 | 0.00 | 0.00 | 0.00 |
| EU-k | 17.62 | 0.61 | 0.11 | 0.11 | 0.33 | 0.58 | 1.83 | 0.47 | 0.00 | 0.00 | 0.00 | 0.00 |
| Bad-T | 22.43 | 0.37 | 10.13 | 0.15 | 1.67 | 1.25 | 4.90 | 2.10 | 1.34 | 1.20 | 0.67 | 0.94 |
| SCRUB | 16.55 | 0.11 | 0.0 | 0.0 | 29.33 | 3.21 | 2.07 | 0.31 | 0.0 | 0.0 | 1.67 | 0.58 |

Table 12: **Large-scale, selective unlearning** results with All-CNN for the **Removing Biases (RB)** application. SCRUB and NegGrad+ are the top-performers in terms of forgetting, though NegGrad+ has worse test performance than SCRUB in both cases. Note also that NegGrad+ isn't as consistent at forgetting across settings as SCRUB, as can be seen in Figure 2

| model | Test error (↓) | | Retain error (↓) | | Forget error (↑) | | IC test error (↓) | | IC retain error (↓) | | Fgt test error (↓) | | Fgt retain error (↓) | |
|---|---|---|---|---|---|---|---|---|---|---|---|---|---|---|
| | mean | std | mean | std | mean | std | mean | std | mean | std | mean | std | mean | std |
| Retrain | 26.67 | 2.87 | 0.0 | 0.0 | 90.33 | 1.53 | 24.0 | 1.8 | 0.0 | 0.0 | 18.33 | 4.16 | 0.0 | 0.0 |
| Original | 41.0 | 2.09 | 0.0 | 0.0 | 0.0 | 0.0 | 56.0 | 3.04 | 0.0 | 0.0 | 92.0 | 7.94 | 0.0 | 0.0 |
| Finetune | 38.13 | 1.42 | 0.0 | 0.0 | 0.0 | 0.0 | 52.0 | 3.12 | 0.0 | 0.0 | 79.33 | 10.07 | 0.0 | 0.0 |
| NegGrad+ | 36.27 | 0.42 | 0.0 | 0.0 | 12.67 | 21.94 | 47.5 | 5.27 | 0.0 | 0.0 | 69.0 | 13.53 | 0.0 | 0.0 |
| CF-k | 39.6 | 1.64 | 0.0 | 0.0 | 0.0 | 0.0 | 54.83 | 2.02 | 0.0 | 0.0 | 85.33 | 7.02 | 0.0 | 0.0 |
| EU-k | 37.47 | 1.62 | 7.33 | 1.26 | 43.67 | 2.08 | 47.0 | 4.77 | 8.33 | 4.73 | 63.33 | 9.71 | 3.67 | 2.52 |
| Fisher | 44.8 | 2.36 | 21.33 | 3.45 | 32.0 | 11.53 | 51.5 | 7.47 | 26.33 | 9.5 | 79.0 | 3.61 | 20.0 | 7.94 |
| NTK | 32.6 | 2.51 | 0.0 | 0.0 | 60.33 | 0.58 | 37.5 | 4.0 | 0.0 | 0.0 | 52.0 | 10.58 | 0.0 | 0.0 |
| SCRUB | 25.93 | 3.13 | 1.08 | 0.52 | 96.0 | 1.73 | 19.0 | 3.91 | 0.0 | 0.0 | 19.67 | 7.51 | 0.0 | 0.0 |

Table 13: Results on CIFAR-5 with ResNet for the **Resolving Confusion (RC)** application. (Confused class 0,1; 50-50 samples). SCRUB is the best-performer by far in terms of eliminating the confusion via unlearning (see the IC error and Fgt error columns), while not hurting performance for other classes (see e.g. the usual Error metrics in the first 3 groups of columns).

## 15 Additional Results for Resolving Confusion (RC)

We show the full results are in Tables 13, 14, 15, 16, 17, 18, 19, 20 for all settings. We observe that across the board, SCRUB is a top-performer on this metric too (see the captions of the individual tables for more details about performance profile).

## 16 Additional results for User Privacy (UP)

We present Basic MIA results for all settings in Tables 21, 22, 23, 24, 25, 26, 27, 28. We find that SCRUB, especially equipped with its rewinding procedure, is able to consistently have a strong defense against MIAs.

| model | Test error (↓) | | Retain error (↓) | | Forget error (↑) | | IC test error (↓) | | IC retain error (↓) | | Fgt test error (↓) | | Fgt retain error (↓) | |
|---|---|---|---|---|---|---|---|---|---|---|---|---|---|---|
| | mean | std | mean | std | mean | std | mean | std | mean | std | mean | std | mean | std |
| Retrain | 24.4 | 2.75 | 0.0 | 0.0 | 90.67 | 4.04 | 19.0 | 1.32 | 0.0 | 0.0 | 11.33 | 4.62 | 0.0 | 0.0 |
| Original | 37.07 | 4.67 | 1.5 | 2.6 | 5.67 | 9.81 | 49.0 | 4.77 | 6.0 | 10.39 | 80.67 | 12.58 | 6.0 | 10.39 |
| Finetune | 34.33 | 3.35 | 0.0 | 0.0 | 3.0 | 5.2 | 43.67 | 7.29 | 0.0 | 0.0 | 67.33 | 16.04 | 0.0 | 0.0 |
| NegGrad+ | 33.53 | 4.47 | 0.0 | 0.0 | 13.33 | 21.36 | 42.33 | 11.34 | 0.0 | 0.0 | 62.0 | 22.65 | 0.0 | 0.0 |
| CF-k | 36.13 | 4.21 | 0.0 | 0.0 | 0.33 | 0.58 | 47.83 | 5.8 | 0.0 | 0.0 | 76.33 | 14.43 | 0.0 | 0.0 |
| EU-k | 51.6 | 1.0 | 27.67 | 3.5 | 52.67 | 6.03 | 59.5 | 5.22 | 38.33 | 6.66 | 68.67 | 15.57 | 19.67 | 10.41 |
| Fisher | 51.93 | 2.95 | 35.17 | 3.92 | 31.0 | 11.53 | 56.83 | 8.69 | 31.67 | 14.01 | 78.33 | 15.53 | 17.67 | 11.5 |
| NTK | 32.2 | 2.84 | 0.75 | 1.3 | 43.33 | 14.15 | 36.67 | 4.07 | 3.0 | 5.2 | 54.33 | 9.02 | 3.0 | 5.2 |
| SCRUB | 25.0 | 3.14 | 0.0 | 0.0 | 93.33 | 2.52 | 26.0 | 4.44 | 0.0 | 0.0 | 18.0 | 11.14 | 0.0 | 0.0 |

Table 14: Results on CIFAR-5 with All-CNN for the **Resolving Confusion (RC)** application. (Confused class 0,1; 50-50 samples). SCRUB is the best-performer by far in terms of eliminating the confusion via unlearning (see the IC error and Fgt error columns), while not hurting performance for other classes (see e.g. the usual Error metrics in the first 3 groups of columns).

| model | Test error (↓) mean | std | Retain error (↓) mean | std | Forget error (↑) mean | std | IC test error (↓) mean | std | IC retain error (↓) mean | std | Fgt test error (↓) mean | std | Fgt retain error (↓) mean | std |
|---|---|---|---|---|---|---|---|---|---|---|---|---|---|---|
| Retrain | 6.0 | 0.2 | 0.0 | 0.0 | 99.67 | 0.58 | 7.17 | 2.57 | 0.0 | 0.0 | 0.0 | 0.0 | 0.0 | 0.0 |
| Original | 27.07 | 3.33 | 1.67 | 0.88 | 4.33 | 1.53 | 57.5 | 6.26 | 6.67 | 3.51 | 108.0 | 14.18 | 6.67 | 3.51 |
| Finetune | 18.8 | 4.26 | 0.0 | 0.0 | 14.67 | 6.03 | 37.67 | 11.15 | 0.0 | 0.0 | 63.67 | 22.01 | 0.0 | 0.0 |
| NegGrad+ | 17.8 | 2.95 | 1.67 | 0.72 | 55.33 | 2.08 | 33.17 | 5.25 | 5.33 | 1.53 | 56.67 | 12.9 | 4.33 | 1.53 |
| CF-k | 22.27 | 4.31 | 0.08 | 0.14 | 10.67 | 5.03 | 46.33 | 10.97 | 0.33 | 0.58 | 81.67 | 23.01 | 0.33 | 0.58 |
| EU-k | 15.27 | 3.19 | 0.83 | 0.38 | 62.0 | 12.49 | 29.33 | 9.0 | 2.33 | 1.53 | 43.67 | 16.29 | 0.33 | 0.58 |
| Fisher | 35.87 | 3.33 | 17.75 | 3.78 | 27.33 | 3.79 | 60.0 | 5.27 | 31.0 | 7.94 | 109.0 | 14.53 | 30.0 | 7.0 |
| NTK | 14.53 | 5.22 | 0.0 | 0.0 | 51.67 | 23.18 | 27.17 | 11.3 | 0.0 | 0.0 | 43.33 | 25.32 | 0.0 | 0.0 |
| SCRUB | 8.47 | 1.17 | 0.33 | 0.14 | 96.0 | 1.0 | 11.33 | 3.82 | 1.33 | 0.58 | 9.33 | 1.53 | 1.33 | 0.58 |

Table 15: Results on Lacuna-5 with ResNet for the **Resolving Confusion (RC)** application. (Confused class 0,1; 50-50 samples). SCRUB is the best-performer by far in terms of eliminating the confusion via unlearning (see the IC error and Fgt error columns), while not hurting performance for other classes (see e.g. the usual Error metrics in the first 3 groups of columns). NTK in some cases is able to resolve confusion, but not consistently, and it also suffers from higher Test Error.

| model | Test error (↓) mean | std | Retain error (↓) mean | std | Forget error (↑) mean | std | IC test error (↓) mean | std | IC retain error (↓) mean | std | Fgt test error (↓) mean | std | Fgt retain error (↓) mean | std |
|---|---|---|---|---|---|---|---|---|---|---|---|---|---|---|
| Retrain | 4.2 | 0.87 | 0.0 | 0.0 | 100.0 | 0.0 | 5.33 | 2.25 | 0.0 | 0.0 | 0.0 | 0.0 | 0.0 | 0.0 |
| Original | 25.47 | 2.32 | 5.75 | 5.63 | 20.33 | 25.74 | 56.17 | 4.93 | 23.0 | 22.54 | 105.67 | 8.08 | 23.0 | 22.54 |
| Finetune | 12.8 | 2.8 | 0.0 | 0.0 | 23.0 | 7.94 | 25.83 | 7.75 | 0.0 | 0.0 | 39.67 | 12.74 | 0.0 | 0.0 |
| NegGrad+ | 12.8 | 9.06 | 2.5 | 3.12 | 90.0 | 6.56 | 20.33 | 17.04 | 5.0 | 6.24 | 12.67 | 11.68 | 2.67 | 3.79 |
| CF-k | 21.27 | 1.63 | 0.58 | 0.8 | 9.33 | 0.58 | 47.0 | 4.58 | 2.33 | 3.21 | 82.67 | 10.12 | 2.33 | 3.21 |
| EU-k | 17.0 | 8.91 | 3.92 | 3.99 | 92.33 | 4.93 | 35.0 | 18.26 | 13.0 | 11.36 | 3.67 | 4.73 | 0.0 | 0.0 |
| Fisher | 49.6 | 4.73 | 39.25 | 7.45 | 40.0 | 9.54 | 57.67 | 10.79 | 42.33 | 11.59 | 88.67 | 11.68 | 29.67 | 16.86 |
| NTK | 12.87 | 6.63 | 2.83 | 4.91 | 72.33 | 12.06 | 25.5 | 17.88 | 11.33 | 19.63 | 35.67 | 24.03 | 10.0 | 17.32 |
| SCRUB | 3.87 | 0.7 | 0.0 | 0.0 | 100.0 | 0.0 | 4.33 | 1.26 | 0.0 | 0.0 | 0.0 | 0.0 | 0.0 | 0.0 |

Table 16: Results on Lacuna-5 with All-CNN for the **Resolving Confusion (RC)** application. (Confused class 0,1; 50-50 samples). SCRUB is the best-performer by far in terms of eliminating the confusion via unlearning (see the IC error and Fgt error columns), while not hurting performance for other classes (see e.g. the usual Error metrics in the first 3 groups of columns).

| model | Test error (↓) mean | std | Retain error (↓) mean | std | Forget error (↑) mean | std | IC test error (↓) mean | std | IC retain error (↓) mean | std | Fgt test error (↓) mean | std | Fgt retain error (↓) mean | std |
|---|---|---|---|---|---|---|---|---|---|---|---|---|---|---|
| retrain | 18.7 | 0.07 | 0.0 | 0.0 | 98.57 | 0.28 | 14.78 | 0.18 | 0.0 | 0.0 | 31.33 | 2.08 | 0.0 | 0.0 |
| original | 21.86 | 0.37 | 0.0 | 0.0 | 0.0 | 0.0 | 31.23 | 0.45 | 0.0 | 0.0 | 356.0 | 11.53 | 0.0 | 0.0 |
| finetune | 20.85 | 0.37 | 0.0 | 0.0 | 0.0 | 0.0 | 26.75 | 0.48 | 0.0 | 0.0 | 255.0 | 10.58 | 0.0 | 0.0 |
| NegGrad+ | 23.41 | 0.32 | 3.87 | 0.31 | 80.07 | 6.77 | 41.08 | 0.6 | 20.29 | 1.52 | 46.0 | 8.72 | 0.67 | 1.15 |
| CF-k | 20.93 | 0.38 | 0.0 | 0.0 | 0.0 | 0.0 | 27.27 | 0.76 | 0.0 | 0.0 | 267.33 | 16.17 | 0.0 | 0.0 |
| EU-k | 20.03 | 0.19 | 0.25 | 0.08 | 95.55 | 0.54 | 17.85 | 0.67 | 0.18 | 0.03 | 53.0 | 7.94 | 3.33 | 2.31 |
| SCRUB | 18.01 | 0.18 | 0.02 | 0.01 | 95.45 | 0.26 | 15.07 | 0.99 | 0.04 | 0.03 | 30.33 | 3.79 | 0.33 | 0.58 |

Table 17: Results on CIFAR-10 with ResNet for the **Resolving Confusion (RC)** application. (Confused class 0,1; 2000-2000 samples).

| model | Test error (↓) mean | std | Retain error (↓) mean | std | Forget error (↑) mean | std | IC test error (↓) mean | std | IC retain error (↓) mean | std | Fgt test error (↓) mean | std | Fgt retain error (↓) mean | std |
|---|---|---|---|---|---|---|---|---|---|---|---|---|---|---|
| retrain | 16.43 | 0.03 | 0.0 | 0.0 | 98.42 | 0.15 | 14.37 | 0.24 | 0.0 | 0.0 | 23.67 | 2.31 | 0.0 | 0.0 |
| original | 19.95 | 0.23 | 0.0 | 0.0 | 0.0 | 0.0 | 30.18 | 0.66 | 0.0 | 0.0 | 348.67 | 13.58 | 0.0 | 0.0 |
| finetune | 18.72 | 0.11 | 0.0 | 0.0 | 1.05 | 0.61 | 24.33 | 0.2 | 0.0 | 0.0 | 223.67 | 6.66 | 0.0 | 0.0 |
| NegGrad+ | 21.74 | 0.44 | 4.48 | 0.34 | 87.65 | 2.98 | 40.05 | 0.44 | 21.8 | 0.66 | 44.0 | 5.2 | 2.33 | 3.21 |
| CF-k | 19.31 | 0.23 | 0.0 | 0.0 | 0.0 | 0.0 | 27.45 | 0.61 | 0.0 | 0.0 | 294.0 | 4.36 | 0.0 | 0.0 |
| EU-k | 17.66 | 0.23 | 1.36 | 0.19 | 87.9 | 1.28 | 16.82 | 0.79 | 2.89 | 0.46 | 63.67 | 8.62 | 91.67 | 12.58 |
| SCRUB | 15.92 | 0.17 | 0.2 | 0.06 | 87.47 | 1.46 | 14.98 | 0.13 | 0.39 | 0.15 | 54.0 | 3.61 | 9.67 | 2.52 |

Table 18: Results on CIFAR-10 with All-CNN for the **Resolving Confusion (RC)** application. (Confused class 0,1; 2000-2000 samples).

| model | Test error (↓) mean | std | Retain error (↓) mean | std | Forget error (↑) mean | std | IC test error (↓) mean | std | IC retain error (↓) mean | std | Fgt test error (↓) mean | std | Fgt retain error (↓) mean | std |
|---|---|---|---|---|---|---|---|---|---|---|---|---|---|---|
| retrain | 2.43 | 0.32 | 0.0 | 0.0 | 99.83 | 0.29 | 3.33 | 0.58 | 0.0 | 0.0 | 0.0 | 0.0 | 0.0 | 0.0 |
| original | 7.37 | 0.31 | 1.21 | 0.12 | 15.83 | 4.19 | 27.67 | 1.26 | 8.26 | 0.8 | 46.0 | 4.36 | 36.33 | 3.51 |
| finetune | 4.17 | 0.5 | 0.0 | 0.0 | 56.83 | 9.44 | 11.17 | 1.76 | 0.0 | 0.0 | 15.67 | 3.51 | 0.0 | 0.0 |
| NegGrad+ | 5.63 | 0.38 | 0.31 | 0.22 | 71.33 | 9.88 | 19.33 | 1.04 | 2.12 | 1.51 | 8.33 | 2.31 | 0.0 | 0.0 |
| CF-k | 5.4 | 0.4 | 0.07 | 0.06 | 33.83 | 3.33 | 17.33 | 1.76 | 0.45 | 0.39 | 27.67 | 3.51 | 2.0 | 1.73 |
| EU-k | 3.0 | 0.26 | 0.0 | 0.0 | 90.17 | 4.65 | 6.0 | 1.8 | 0.0 | 0.0 | 2.0 | 2.65 | 0.0 | 0.0 |
| SCRUB | 3.07 | 0.59 | 0.0 | 0.0 | 98.5 | 0.5 | 6.83 | 1.26 | 0.0 | 0.0 | 0.67 | 0.58 | 0.0 | 0.0 |

Table 19: Results on Lacuna-10 with ResNet for the **Resolving Confusion (RC)** application. (Confused class 0,1; 200-200 samples).

| model | Test error (↓) mean | std | Retain error (↓) mean | std | Forget error (↑) mean | std | IC test error (↓) mean | std | IC retain error (↓) mean | std | Fgt test error (↓) mean | std | Fgt retain error (↓) mean | std |
|---|---|---|---|---|---|---|---|---|---|---|---|---|---|---|
| retrain | 2.13 | 0.25 | 0.0 | 0.0 | 99.83 | 0.29 | 2.5 | 1.32 | 0.0 | 0.0 | 0.0 | 0.0 | 0.0 | 0.0 |
| original | 7.83 | 0.55 | 1.21 | 0.52 | 16.0 | 4.92 | 31.33 | 1.76 | 8.26 | 3.53 | 56.33 | 3.79 | 36.33 | 15.53 |
| finetune | 3.0 | 0.7 | 0.0 | 0.0 | 74.5 | 6.08 | 6.5 | 2.0 | 0.0 | 0.0 | 9.0 | 3.0 | 0.0 | 0.0 |
| NegGrad+ | 4.3 | 0.52 | 0.4 | 0.06 | 89.67 | 4.25 | 15.33 | 2.47 | 2.73 | 0.39 | 4.67 | 3.21 | 0.0 | 0.0 |
| CF-k | 5.27 | 0.47 | 0.11 | 0.07 | 33.33 | 1.61 | 18.5 | 2.29 | 0.76 | 0.47 | 31.33 | 3.51 | 3.33 | 2.08 |
| EU-k | 2.53 | 0.67 | 0.09 | 0.02 | 97.83 | 2.08 | 5.17 | 1.04 | 0.38 | 0.35 | 0.33 | 0.58 | 0.67 | 0.58 |
| SCRUB | 2.1 | 0.4 | 0.0 | 0.0 | 97.5 | 1.73 | 4.17 | 0.58 | 0.0 | 0.0 | 0.33 | 0.58 | 0.0 | 0.0 |

Table 20: Results on Lacuna-10 with All-CNN for the **Resolving Confusion (RC)** application. (Confused class 0,1; 200-200 samples).

| method | Test error mean | std | Forget error mean | std | Retain error mean | std | MIA mean | std |
|---|---|---|---|---|---|---|---|---|
| Retrain | 16.71 | 0.05 | 26.67 | 3.09 | 0.00 | 0.00 | 51.33 | 6.13 |
| Original | 16.71 | 0.07 | 0.00 | 0.00 | 0.00 | 0.00 | 68.67 | 3.09 |
| Finetune | 16.86 | 0.13 | 0.00 | 0.00 | 0.00 | 0.00 | 69.33 | 2.05 |
| NegGrad+ | 21.65 | 0.40 | 47.00 | 3.74 | 4.54 | 0.70 | 73.00 | 1.41 |
| CF-k | 16.82 | 0.03 | 0.00 | 0.00 | 0.00 | 0.00 | 69.67 | 1.89 |
| EU-k | 18.44 | 0.21 | 0.33 | 0.47 | 0.32 | 0.02 | 66.00 | 2.94 |
| Bad-T | 22.43 | 0.37 | 1.67 | 1.25 | 10.13 | 0.15 | 77.67 | 4.11 |
| SCRUB | 17.01 | 0.20 | 33.00 | 5.89 | 0.00 | 0.00 | 51.00 | 1.41 |
| SCRUB+R | 16.88 | 0.19 | 26.33 | 4.50 | 0.00 | 0.00 | 49.33 | 2.49 |

Table 21: Basic MIA for All-CNN architecture on CIFAR-10 for selective unlearning, for the **User Privacy (UP)** application.

| method | Test error mean | std | Forget error mean | std | Retain error mean | std | MIA mean | std |
|---|---|---|---|---|---|---|---|---|
| Retrain | 13.98 | 0.07 | 100.00 | 0.00 | 0.00 | 0.00 | 48.73 | 0.24 |
| Original | 15.70 | 0.09 | 0.00 | 0.00 | 0.00 | 0.00 | 71.40 | 0.70 |
| Finetune | 14.53 | 0.13 | 1.31 | 0.54 | 0.00 | 0.00 | 74.97 | 1.27 |
| NegGrad+ | 17.04 | 0.11 | 59.91 | 1.53 | 0.43 | 0.09 | 70.03 | 1.92 |
| CF-k | 15.72 | 0.06 | 0.00 | 0.00 | 0.00 | 0.00 | 72.93 | 1.06 |
| EU-k | 15.76 | 0.28 | 100.00 | 0.00 | 0.24 | 0.02 | 51.60 | 1.22 |
| Bad-T | 16.98 | 0.40 | 81.93 | 3.50 | 5.84 | 0.43 | 58.07 | 1.76 |
| SCRUB | 14.93 | 0.17 | 100.00 | 0.00 | 0.09 | 0.02 | 54.30 | 2.24 |
| SCRUB+R | 14.93 | 0.17 | 100.00 | 0.00 | 0.09 | 0.02 | 54.30 | 2.24 |

Table 22: Basic MIA for All-CNN architecture on CIFAR-10 for class unlearning, for the **User Privacy (UP)** application.

| method | Test error | | Forget error | | Retain error | | MIA | |
|---|---|---|---|---|---|---|---|---|
| | mean | std | mean | std | mean | std | mean | std |
| Retrain | 17.38 | 0.15 | 29.33 | 2.49 | 0.00 | 0.00 | 54.00 | 1.63 |
| Original | 17.41 | 0.15 | 0.00 | 0.00 | 0.00 | 0.00 | 65.33 | 0.47 |
| Finetune | 17.48 | 0.16 | 0.00 | 0.00 | 0.00 | 0.00 | 64.00 | 0.82 |
| NegGrad+ | 21.69 | 0.07 | 45.33 | 2.62 | 3.94 | 0.43 | 66.67 | 1.70 |
| CF-k | 17.53 | 0.19 | 0.00 | 0.00 | 0.00 | 0.00 | 65.00 | 0.00 |
| EU-k | 19.77 | 0.04 | 13.67 | 0.47 | 0.06 | 0.01 | 53.00 | 3.27 |
| Bad-T | 23.47 | 1.57 | 34.67 | 1.70 | 14.53 | 1.65 | 59.67 | 4.19 |
| SCRUB | 17.01 | 0.03 | 71.67 | 0.94 | 0.01 | 0.01 | 78.00 | 2.45 |
| SCRUB+R | 17.54 | 0.28 | 19.33 | 14.64 | 0.01 | 0.01 | 58.67 | 1.89 |

Table 23: Basic MIA for ResNet architecture on CIFAR-10 for selective unlearning, for the **User Privacy (UP)** application.

| method | Test error | | Forget error | | Retain error | | MIA | |
|---|---|---|---|---|---|---|---|---|
| | mean | std | mean | std | mean | std | mean | std |
| Retrain | 14.69 | 0.10 | 100.00 | 0.00 | 0.00 | 0.00 | 49.33 | 1.67 |
| Original | 16.33 | 0.14 | 0.00 | 0.00 | 0.00 | 0.00 | 71.10 | 0.67 |
| Finetune | 15.10 | 0.16 | 0.33 | 0.17 | 0.00 | 0.00 | 75.57 | 0.69 |
| NegGrad+ | 17.41 | 0.09 | 61.00 | 1.14 | 0.44 | 0.05 | 69.57 | 1.19 |
| CF-k | 15.29 | 0.02 | 0.04 | 0.04 | 0.00 | 0.00 | 75.73 | 0.34 |
| EU-k | 17.05 | 0.07 | 97.48 | 0.28 | 0.05 | 0.01 | 54.20 | 2.27 |
| Bad-T | 19.56 | 1.44 | 11.34 | 1.82 | 94.67 | 6.12 | 54.33 | 0.31 |
| SCRUB | 15.33 | 0.06 | 100.00 | 0.00 | 0.08 | 0.01 | 52.20 | 1.71 |
| SCRUB+R | 15.33 | 0.06 | 100.00 | 0.00 | 0.08 | 0.01 | 52.20 | 1.71 |

Table 24: Basic MIA for ResNet architecture on CIFAR-10 for class unlearning, for the **User Privacy (UP)** application.

| method | Test error | | Forget error | | Retain error | | MIA | |
|---|---|---|---|---|---|---|---|---|
| | mean | std | mean | std | mean | std | mean | std |
| Retrain | 1.50 | 0.08 | 0.33 | 0.47 | 0.00 | 0.00 | 52.00 | 2.16 |
| Original | 1.57 | 0.24 | 0.00 | 0.00 | 0.00 | 0.00 | 59.00 | 2.16 |
| Finetune | 1.40 | 0.16 | 0.00 | 0.00 | 0.00 | 0.00 | 57.33 | 3.30 |
| NegGrad+ | 3.60 | 0.14 | 14.33 | 1.25 | 0.87 | 0.07 | 51.00 | 1.63 |
| CF-k | 1.57 | 0.12 | 0.00 | 0.00 | 0.00 | 0.00 | 58.33 | 2.49 |
| EU-k | 3.90 | 1.47 | 0.00 | 0.00 | 0.76 | 0.63 | 52.00 | 3.56 |
| Bad-T | 4.90 | 2.10 | 1.34 | 1.20 | 0.67 | 0.94 | 67.67 | 6.94 |
| SCRUB | 1.67 | 0.19 | 0.00 | 0.00 | 0.00 | 0.00 | 57.67 | 0.94 |
| SCRUB+R | 1.67 | 0.19 | 0.00 | 0.00 | 0.00 | 0.00 | 57.67 | 0.94 |

Table 25: Basic MIA for All-CNN architecture on Lacuna-10 for selective unlearning, for the **User Privacy (UP)** application.

| method | Test error | | Forget error | | Retain error | | MIA | |
|---|---|---|---|---|---|---|---|---|
| | mean | std | mean | std | mean | std | mean | std |
| Retrain | 1.67 | 0.09 | 100.00 | 0.00 | 0.00 | 0.00 | 55.67 | 2.62 |
| Original | 1.70 | 0.21 | 0.00 | 0.00 | 0.00 | 0.00 | 58.00 | 1.63 |
| Finetune | 1.67 | 0.27 | 0.00 | 0.00 | 0.00 | 0.00 | 56.33 | 1.25 |
| NegGrad+ | 2.00 | 0.00 | 14.27 | 0.74 | 0.00 | 0.00 | 54.33 | 2.05 |
| CF-k | 2.07 | 0.14 | 0.00 | 0.00 | 0.00 | 0.00 | 52.33 | 2.05 |
| EU-k | 4.15 | 1.22 | 62.08 | 44.26 | 0.81 | 0.53 | 52.67 | 3.68 |
| Bad-T | 2.56 | 0.09 | 38.65 | 36.80 | 0.37 | 0.18 | 63.33 | 2.49 |
| SCRUB | 1.96 | 0.34 | 100.00 | 0.00 | 0.00 | 0.00 | 50.33 | 2.62 |
| SCRUB+R | 1.96 | 0.34 | 100.00 | 0.00 | 0.00 | 0.00 | 50.33 | 2.62 |

Table 26: Basic MIA for All-CNN architecture on Lacuna-10 for class unlearning, for the **User Privacy (UP)** application.

| method | Test error | | Forget error | | Retain error | | MIA | |
|---|---|---|---|---|---|---|---|---|
| | mean | std | mean | std | mean | std | mean | std |
| Retrain | 2.50 | 0.24 | 1.67 | 0.94 | 0.00 | 0.00 | 49.67 | 3.09 |
| Original | 2.53 | 0.25 | 0.00 | 0.00 | 0.00 | 0.00 | 56.67 | 1.70 |
| Finetune | 2.67 | 0.05 | 0.00 | 0.00 | 0.00 | 0.00 | 53.67 | 0.94 |
| NegGrad+ | 4.30 | 0.43 | 12.67 | 3.30 | 0.95 | 0.08 | 54.00 | 2.16 |
| CF-k | 2.47 | 0.25 | 0.00 | 0.00 | 0.00 | 0.00 | 56.00 | 0.82 |
| EU-k | 2.60 | 0.00 | 0.00 | 0.00 | 0.03 | 0.00 | 56.00 | 2.83 |
| Bad-T | 7.30 | 2.20 | 3.26 | 1.83 | 0.33 | 0.47 | 67.33 | 3.40 |
| SCRUB | 2.97 | 0.25 | 6.00 | 3.27 | 0.00 | 0.00 | 50.67 | 4.03 |
| SCRUB+R | 2.97 | 0.25 | 6.00 | 3.27 | 0.00 | 0.00 | 50.67 | 4.03 |

Table 27: Basic MIA for ResNet architecture on Lacuna-10 for selective unlearning, for the **User Privacy (UP)** application.

| method | Test error | | Forget error | | Retain error | | MIA | |
|---|---|---|---|---|---|---|---|---|
| | mean | std | mean | std | mean | std | mean | std |
| Retrain | 2.52 | 0.19 | 100.00 | 0.00 | 0.00 | 0.00 | 55.00 | 2.94 |
| Original | 2.81 | 0.28 | 0.00 | 0.00 | 0.00 | 0.00 | 56.00 | 2.45 |
| Finetune | 3.04 | 0.19 | 0.00 | 0.00 | 0.00 | 0.00 | 54.67 | 1.25 |
| NegGrad+ | 2.74 | 0.26 | 9.48 | 0.64 | 0.00 | 0.00 | 53.67 | 4.03 |
| CF-k | 2.81 | 0.28 | 0.00 | 0.00 | 0.00 | 0.00 | 56.00 | 2.45 |
| EU-k | 2.48 | 0.14 | 7.71 | 2.52 | 0.00 | 0.00 | 54.33 | 3.09 |
| Bad-T | 3.37 | 0.50 | 67.60 | 24.26 | 1.06 | 0.47 | 58.00 | 2.94 |
| SCRUB | 3.26 | 0.38 | 99.90 | 0.15 | 0.07 | 0.05 | 54.33 | 2.49 |
| SCRUB+R | 3.26 | 0.38 | 99.90 | 0.15 | 0.07 | 0.05 | 54.33 | 2.49 |

Table 28: Basic MIA for ResNet architecture on Lacuna-10 for class unlearning, for the **User Privacy (UP)** application.

