# OpenReview forum: "Towards Unbounded Machine Unlearning"
_NeurIPS.cc/2023/Conference — NeurIPS 2023 poster_

### Official Review · Reviewer_YAUt · 2023-07-03

**Soundness:** 3 good
**Presentation:** 3 good
**Contribution:** 2 fair
**Rating:** 5
**Confidence:** 4

**Summary:**

This paper studies machine unlearning from an empirical perspective, proposes a scalable unlearning method SCRUB based on knowledge distillation, and evaluates the performance under different metrics along with other available unlearning approaches. Overall, this paper may provide insights and empirical metrics for designing other scalable unlearning methods in practice.

**Strengths:**

Firstly, unlike other unlearning papers focusing primarily on user privacy, this paper summarizes three possible applications (removing biases, resolving confusion, and protecting user privacy) for unlearning algorithms to work in practice. Indeed, one should consider different metrics to evaluate the performance of unlearning algorithms under different scenarios. This is a valuable observation for designing other unlearning methods. The authors also show that previous state-of-the-art methods fail to obtain good performance on all tasks, which is exactly due to the fact that previous methods only focus on one criterion and thus cannot have universal good performance.

Secondly, the proposed method SCRUB is based on a teacher-student model formulation, which has high scalability that can support large forget sets. Many previous unlearning methods, especially those having theoretical guarantees, would need some kind of sequentially unlearning process and repeatedly computing the Hessian matrix or Fisher information which leads to unavoidable high computational complexity. On the other hand, SCRUB treats the forget set as a whole and aims to have a consistent performance on the forget set.

Lastly, this paper provides a comprehensive empirical evaluation to validate the superior performance of SCRUB. Moreover, for user privacy applications, it uses the latest LiRA membership inference attack accuracy as a metric and adapts the SCRUB method with a rewinding step, which has indeed proven to be helpful by simulations. The full code and documents are included in the SI for reproducibility.

Overall this paper is well-written and easy to follow.

**Weaknesses:**

Firstly, as the authors have pointed out in the paper, the idea of using knowledge distillation to achieve unlearning is not new, and the metrics, including using MIA accuracy to evaluate the unlearning performance are also not new. So my biggest concern for this paper is that the contribution might seem incremental. Also for the adaptation of the LiRA attacker, although the authors claim that "the first adaptation of LiRA for unlearning", but it is not clear why this type of adaptation is really needed. Is it because this adaptation can greatly increase the attack accuracy? Or is it technically challenging to apply LiRA directly within the framework? I would assume it is the first one, but some clarifications on these would make the paper clearer.

Secondly, I am not sure if SCRUB can work well when the forget set is small but the training set is large. Consider the case where a model uses millions of data points for training, but we only want to unlearn one (or a very small portion) data point. This can happen frequently, especially in the UP case, where only a few users want to withdraw their data at the same time. Based on SCRUB+R, we would need to create a validation set of the same distribution as the forget set, which is basically sample another point from the same class as the validation set, and the goal is to make the model performance on forget point and validation point as close as possible. Since the choice of validation point could be any point from that class, such kind of requirement might lead to unstable training and the final model performance would largely depend on which point we choose. This is just an extreme example, but such kind of trend also appears in Figure 3, where the difference between SCRUB+R and Retrain becomes more significant when the forget set is smaller.

However, despite these weaknesses exist, I think the observations, applications, and evaluation metrics proposed in the paper is still meaning for the unlearning community. So I would recommend boardline accept for the paper.

**Questions:**

Here I summarize some questions from the weakness section.

1. Why the adaptation of LiRA is needed? Is it because allowing the attacker to know the unlearning algorithm can greatly increase the attack accuracy?

2. What would change in the unlearning algorithm when we only want to unlearn a very small portion of the training set?

3. What is the real difference between Removing Biases (RB) and Resolving Confusion (RC)? From the description, it seems that their goals are both not to predict the original labels of the forget set, and they share the same algorithm. I believe they can be both viewed as data cleaning for model training.

**Limitations:**

The authors have listed detailed limitations in the Appendix. First and the most important limitation would be it is hard to analyze SCRUB theoretically, as it always is in deep learning. However, when it comes to user privacy, sometimes a theoretical guarantee is required, which prevents the usage of SCRUB. The authors also mention the difficulty in training the min-max objective function, and how it will affect the model performance. Finally, the authors show cases where the rewinding procedure fails to prevent the MIA model to recognize the forget set, leading to privacy leakage in such cases.

---

> ### Author Rebuttal · Authors · 2023-08-08
>
> We thank reviewer YAUt for their insightful comments.
>
> **RE: incremental contribution**
> Please see our response in the general rebuttal for similarity against other methods.
>
> **RE: Smaller forget-sets and validation distribution**
>
> Thank you for this comment. We should have clarified better and we will in the revised paper. Please note that we’re not restricted to picking a few validation examples if the forget set is small. We’re free to pick as many validation examples as we like (assuming availability, of course), and, as the reviewer points out, we will pick sufficiently many to avoid high-variance issues. The requirement that we have is to have a similar distribution between the validation and forget sets. In the context of classification problems, we simply do this via inspecting the class labels. So, for example, if the forget set contains only cats, we will pick only cats for the validation set too.
> In addition, we have now performed additional experiments where the forget set size is much smaller than those reported in the paper. We used a larger dataset/network (namely, a 100-way classification task using CIFAR-100 and the VGG+BN network, with ~138M parameters). The results shown in Table 1 to Table 3 (please see the tables in our general response, above) clearly show that SCRUB has retain, test, and forget errors very close to retrain-from-scratch, even when the forget set sizes are just 10/40,000, 20/40,000, and 400/40,000 (that is, 0.1%, 0.2%, and 1% of the dataset size) - in addition to the results for 100/40,000 in the paper, showing robustness across forget set sizes)!
>
> **RE: "Why the adaptation of LiRA is needed?"**
>
> Great question! Let us explain. There has been an extensive literature of MIAs (LiRA is the state-of-the-art) *outside of the context of unlearning*. The goal of the attacker there is to infer whether a given example was part of the training set of the model that is under attack. Now, the evaluation protocol in the context of unlearning is different: we don’t care to attack the original model in this case, and we don’t care to attack any example. Instead, we aim to use MIA tools to gauge unlearning quality: we want to see if the unlearned model (the original model, after having undergone an unlearning procedure) is vulnerable to attacks w.r.t the forget examples. That is, to estimate the quality of unlearning, what we care about is to measure whether the forget-set examples are indistinguishable from held-out examples, in the context of the unlearned model.
> Therefore, to build the strongest possible attack in this setting, we take exactly this setup into consideration when constructing shadow models for the attacker. One could of course try applying MIAs out-of-the-box but we suspect (and found in preliminary results in earlier stages of this research project) that those attacks will be generally less successful.
> Overall, the study of MIAs as a tool for evaluating unlearning quality is an open research area (as our privacy-expert colleagues assure us). In our paper, we actually include two different types of MIAs, (one LiRA-like and one not) to show the resilience of SCRUB/SCRUB+R under all these metrics. We sincerely hope that this will be appreciated in terms of (i) the comprehensiveness of the evaluation and (ii) in terms of the consistently strong-performing methods we contribute! As an aside, we strongly believe that our proposed adaptation of LiRA is a fundamental step in that direction (and a significant contribution of its own accord).
>
> **RE: "What would change in the unlearning algorithm when we only want to unlearn a very small portion of the training set"**
>
> Great question! We believe that an extensive experimentation of different unlearning algorithms with different types of forget sets, and different forget set sizes would be a very valuable contribution. However, such an investigation is outside the scope of our paper. We hope the reviewer agrees that our experimentation is already very thorough and well-substantiates our claim that SCRUB is a consistently top-performing algorithm.
>
> Nonetheless, to address this fair comment by the reviewer, we have performed additional experiments. We used a larger dataset/network (namely, a 100-way classification task using CIFAR-100 and the VGG+BN network, with ~138M parameters). The results shown in Table 1 to Table 3 (please see the tables in our general response, above) clearly show that SCRUB has retain, test, and forget errors very close to retrain-from-scratch, even when the forget set sizes are just 10/40,000, 20/40,000, and 400/40,000 (that is, 0.1%, 0.2%, and 1% of the dataset size) - in addition to the results for 100/40,000 in the paper, showing robustness across forget set sizes)!
>
> **RE: "What is the real difference between Removing Biases (RB) and Resolving Confusion (RC)?"**
>
> This is another great point. Yes, to a large extent, the reviewer is correct. However, for the RC case, one can (and we argue that one should) employ application-specific metrics (additionally to retain-, forget-, and test-errors) that address the ability of the model to specifically resolve class mislabeling (confusion). And, for the RC cases, the forget set includes mislabelled examples, whereas for the RB cases, the forget set consists of examples causing biases.
>
>
> We hope the above responses address all key concerns and we are very keen to engage in order to further clarify/address comments and questions!

---

> > ### Comment · Reviewer_YAUt · 2023-08-13
> > **Reviewer Response**
> >
> > I want to thank the authors for preparing the detailed responses and additional experiments. It is interesting to learn that the adaption of LiRA is critical in improving attack accuracy. The authors may want to emphasize this in the main text as well if the space allows.
> >
> > The authors have fully answered my questions and I do not have further questions.

---

> > > ### Author Response · Authors · 2023-08-17
> > > **thank you for the response!**
> > >
> > > Hi reviewer YAUt,
> > >
> > > Thank you for reading our rebuttal, it is great to hear that we have fully answered your questions. We agree with your suggestion and will emphasize the motivation and importance of adapting LiRA to the unlearning protocol.
> > > Given this, we were wondering if you would consider increasing your score? If there are any other concerns remaining we would be very happy to discuss further.
> > >
> > > Thank you again!

---

### Official Review · Reviewer_LTMH · 2023-07-06

**Soundness:** 3 good
**Presentation:** 1 poor
**Contribution:** 3 good
**Rating:** 5
**Confidence:** 3

**Summary:**

The paper proposes three distinct goals to machine learning: removing bias, resolving class confusion from mislabelling of training data, and privacy through the right to be forgotten, and suggests that these goals require different technical approaches. The paper proposes an linearized multi-objective optimization problem (minimize loss on retained examples, minimize average KL divergence between softmax outputs to the original ones on retained examples, and maximize it on forget examples) with an alternating step optimizer. To limit the membership inference risks associated with ensuing high loss on forget examples, the paper proposes early stopping with a specially crafted validation set. The paper evaluates the algorithm for three goals on two datasets and using two network architectures.

**Strengths:**

- S1. The proposed optimization objective is simple and seems to be more effective than a multitude of more complex approaches from prior work.
- S2. Relatedly, the performance of the algorithm seems to be significantly more effective with multiple goals in mind.
- S3. The paper delinates three different goals for unlearning that could need different technical approaches.

**Weaknesses:**

- W1. *Impossible to evaluate the technical contributions without a detailed reading of the appendix.* The main body of the paper does not contain sufficient technical details to understand the contributions, deferring many crucial details to the appendix: the alternating step algorithm which is crucial to the contribution, training dynamics of the proposed objective that is inherent to the proposed optimization objective, the exact construction of the validation set for "rewinding", etc. Although the paper does not read bad, this is the reason I give the lowest score to the presentation rubric. It is not possible to judge the technical contributions without a reading of the appendix, which goes against the point of the page limit.
- W2. *Insufficient evaluation in some aspects.* Although in many regards, the evaluation is comprehensive, e.g., with respect to the coverage of the methods from prior work, it is rather incomplete in other aspects. In particular:
   - We only see the results on two datasets and two models.
   - We do not necessarily know how significant is the performance improvement under optimization uncertainty. Such uncertainty might have non-trivial effect due to the unstable nature of the training dynamics of the proposed optimization objective.
   - The proposed optimization objective is simple and is only slightly different from the prior NegGrad work. I do not think this constitutes lack of novelty, especially as it seems that the proposed simple approach outperforms many complex ones. However, in this case, it is important to see ablation studies with respect to the design choices of this objective. For instance, why KL divergence in softmax, not loss for both retain and forget sets, or for one of the two sets? Are there any possible stability inducing terms that would get rid of periodic training dynamics?
   - Although I understand this is computationally expensive, especially if comparing to the retraining baseline, it would be good to see the performance in at least one large-scale dataset setting. It is important as machine unlearning seems most useful in such settings; otherwise, retraining might be good enough. In choosing the evaluation settings the paper cites prior work, but some of the prior work has evaluated, e.g., on Imagenet-1K.

**Questions:**

---

**Limitations:**

See W2.

---

> ### Author Rebuttal · Authors · 2023-08-08
>
> We thank reviewer LTMH for raising the important points.
>
>
> **RE: Weaknesses, W1**
>
> Thank you for this fair comment. However, we wish to point out that the “scale” of this contribution is very large, vis-a-vis the page limit of NeurIPS submissions. This is, by the way, a common problem for many NeurIPS papers. We tried hard to convey all essential elements of our methods and problem setting and comprehensive experiments and results. Nonetheless,  we will revise the paper write-up, incorporating many of the above key elements in the paper properly.
>
> **RE: "The proposed optimization objective is simple and is only slightly different from the prior NegGrad work"**
>
> Thank you very much for raising this point, as it is critical. We would like to clarify that *the NegGrad baseline we use in this work is a novel unlearning algorithm in its own right* – also our contribution. To make this clearer, we will rename it to NegGrad+ (or something similar). A similar NegGrad baseline has been used in previous works that simply performs gradient ascent on the forget set. We found this to be very poor, so we won’t report it. That observation, though, inspired us to build an improved version of NegGrad that shares structural similarity with SCRUB (due to balancing two terms; one on the retain and one of the forget set). Our motivation for constructing this additional baseline is twofold: 1) it allows us to measure the added gain from SCRUB’s additional complexity *over our improved NegGrad*, and 2) contribute another baseline to the field, which we show is a particularly strong one, and we hope future work reports results with it too. From a pedagogical perspective, we are hoping that reviewers and readers will notice the similarity and then be able to identify/appreciate the added benefits coming with SCRUB/SCRUB+R.
>
> **RE: More/Larger datasets and models**
>
> Thank you for this meaningful comment as well, as indeed in small-scales perhaps retrain-from-scratch is viable. In fact, we have already performed experiments on a larger dataset/network (namely, a 100-way classification task using CIFAR-100 and the VGG+BN network, with ~138M parameters). The results shown in Table 1 to Table 3 (please see the tables in our general response, above) clearly show that SCRUB has retain, test, and forget errors very close to retrain-from-scratch!
>
> **RE: Optimization's uncertainty**
>
> This is a great point. Indeed the min-max optimization of SCRUB may lead to increased variance due to the oscillating behaviour. We have acknowledged this fact as a limitation of SCRUB in the appendix. However, there are two key points to consider: Firstly, we've demonstrated through practical experiments that we can mitigate the impact of this uncertainty by following the training procedures outlined in Algorithm 1 and Algorithm 2. You can find related ablation results in Figure 6 to Figure 9. Secondly, in all the experiments detailed in the paper, we've taken a thorough approach to comparison. We have run each method multiple times using different random seeds and compared their results using 95% confidence intervals. These multiple runs inherently capture some of the uncertainty arising from the optimization process itself. Importantly, the results reveal that the width of SCRUB's 95% confidence interval is not broader when compared to other baseline methods. This suggests that in practice SCRUB in fact does not introduce additional uncertainty beyond what is already present in the underlying setting. And, we find that SCRUB is by far the most consistent top performer across applications, metrics, and architectures out of all considered baselines.
> We leave it to future work to propose more stable optimization algorithms for min-max optimization problems like SCRUB’s but in the meantime, we hope that the reviewer appreciates that we have performed a very thorough investigation that reveals that SCRUB is a top-performing unlearning method across the board. We believe that this is a valuable contribution to the community.
>
> **RE: Ablation studies**
>
> In fact, we have conducted sensitivity analysis against alpha and gamma parameters that show the sensitivity of scrub’s performance w.r.t those parameters, Please see Fig 5 in the appendix (supplementary document), which shows that SCRUB is not very sensitive to these hyperparameters. Additionally, we have ablated the cross-entropy loss in our formulation, please see Fig 10 in the appendix.
> Furthermore, it is worth noting that certain modifications to the terms in our loss formulation will result in some of the other baselines. For instance, using cross-entropy only for the retain-set would yield the ‘Fine-tune’ baseline. Using cross-entropy only for the forget-set yields the simple NegGrad in the existing literature, that tends to catastrophically forget everything, and replacing both KL-divergence terms by cross-entropy recovers our improved proposed NegGrad. Therefore, we can view those baselines as constituting different ablations / variations of SCRUB's objective function.
>
> We hope the above responses address all key concerns and we are very keen to engage in order to further clarify/address comments and questions!

---

> > ### Comment · Reviewer_LTMH · 2023-08-21
> >
> > Apologies for the late response. Given the author's responses, I think the issue with insufficient evaluation/motivation of the objectives is resolved. Because of this, and because of the strong performance results, I am increasing my score.

---

### Official Review · Reviewer_Xo5D · 2023-07-08

**Soundness:** 3 good
**Presentation:** 2 fair
**Contribution:** 3 good
**Rating:** 6
**Confidence:** 5

**Summary:**

The paper presents a Machine Unlearning method and shows its utility in Removing bias, Resolving confusion along with protecting user privacy. The experiments are conducted with different sets of evaluation metrics for each application. The latest Membership Inference Attack is adopted for Unlearning to evaluate the information leak in the unlearned model. The paper also stresses on relaxing the requirements of theoretical proofs for the bounds on the Machine Unlearning algorithms in general. This argued is due to the Unlearning methods utility in other applications than Data Privacy in ML.

**Strengths:**

The following are some of the strength of the paper:
1. The paper shows the utility of Machine Unlearning in removing Bias and Resolving confusion.
2. The paper is well written and easily understandable.
3. Adoption of LiRA MIA attack is a good addition and to the evaluation of Unlearned Model.
4. Several experiments are conducted and comparison with some of the existing methods is discussed.

**Weaknesses:**

1. Excessive Similarity with an existing paper – The main idea of this paper shares a lot of similarity with the following paper https://ojs.aaai.org/index.php/AAAI/article/view/25879/25651 (mentioned as Bad-T in the paper)

In Page-6, Line 239-241, it is mentioned that “further differences with Bad-T is discussed in Appendix”. However, I could not find any such comparison. Since, there is a lot of similarity between the proposed method and Bad-T, it would have been appropriate to give a technical difference between the proposed and Bad-T method. But only the experimental results are compared in the Appendix.

There are excessive similarities in the method described in Section 3.1 and Bad-T paper. The list of the similarities are:

a. Page 3, Line 129-136. This is same as the Bad-T paper which also treats the original model as a teacher and the student selectively obeys the teacher for retain (Dr) and forget (Df) samples. Refer to the “Section: Proposed Method” in Bad-T Paper. The training object in Bad-T is also based on the KL-Divergence between the output distributions of the teacher and student model.

b. Similar to Bad-T method, the teacher model is frozen here. Similar to Bad-T paper, the student is initialized with the weights of the teacher. Only difference is that the optimization is done alternatively for the retain (Dr)  and forget (Df) samples. This is again quite similar to Bad-T optimization where the optimization for retain (Dr) samples are done using the competent teacher (original model) and optimization for the forget (Df) samples are done using the bad teacher which acts as a good proxy for the student model to move away from the original model. Particularly, if we take the variation of Bad-T where the bad teacher is just a random generator, wouldn’t that give a better robustness against MIA attacks? Because it is training the model predict random outputs instead of deliberately attempting to move the weights towards the opposite direction for forget samples (Df).

c. The proposed method might be more prone to being unstable as already discussed in the limitation section. A comparison with Bad-T in terms of Stability was desired.

d. Overall, it seems the entire method is almost similar to the Bad-T paper. Besides, the authors did not clearly mention the technical difference with Bad-T. Furthermore, as mentioned in the paper, Bad-T is significantly faster than the proposed method. In Machine Unlearning, speed is one of the most important factor. It is quite possible that the Bad-T method (or its other variants) might give similar results on all the experiments presented in the paper with better speed.

2. Inadequate Literature Study:

In Page 2, Line 54-58, the authors explains that there is a lack of Machine Unlearning methods that are free from the assumption of the stability of SGD and can work with bigger datasets. In the literature study Section (Page 5) also, the authors primarily highlight the 3 papers by Golatkar et al. and derive their motivation for this paper.

In past few years, there have been several papers that have highlighted the impracticality, inefficiency, and non-scalability issues present in the methods by Golatkar et al. Some of the papers that have shown Unlearning results without relying on the stability of SGD and other assumptions in Golatkar et al.:
[1] “Amnesiac Machine Learning”
[2] “Can Bad Teaching Induce Forgetting? Unlearning in Deep Networks Using an Incompetent Teacher”
[3] “Fast Yet Effective Machine Unlearning”
Some more papers can be found here: https://github.com/jjbrophy47/machine_unlearning

The idea of ‘unbound’ is not new as claimed in this paper. All these 3 papers have in fact shown good unlearning in ‘unbound’ setup. As far scalability is concerned, [3] have shown results on CIFAR-100 and VGGFACE-100 (class-level unlearning). The paper fail to mention these salient points in the literature study and in the experiments.

3. Considering the above mentioned points, the in the title of Section 2, the word ‘Unbound’ appears trivial. This Section gives a generic definition and discusses existing trade-offs in Machine Unlearning (MU). The content in this section doesn’t add anything significant to define ‘Unbound’. In fact, the Section title is little confusing. For example, in case of Privacy, we would like the MU method to be bound? Or in general unbounded methods are fine as they can be used to address multiple type of problems?

4. About SCRUB+R: Maybe this could have been part of the experiments and analysis section? It is just Early Stopping….

**Questions:**

1. The authors could let me know their thoughts on Weakness-1 and Weakness-2.

2. Since, several existing Machine Unlearning methods have been presented in the ‘unbound’ setting (refer to weakness section). Maybe, it would be nice to discuss the idea of ‘unbound’ as a general category of methods and not something novel presented in this paper?

3. In Bad-T, there is a metric called Zero Retrain Forgetting (ZRF) which also aims to measure the generalization power of the unlearned model and not focus just on the Df and Dr accuracies. Is there any way the ZRF metric could be incorporated in this method?

4. I have few doubts about Removing Biases experiments – Are you unlearning a class or a subset of class or a set of random samples here? Also, it is unclear to me how the experiments on Removing biases are different from standard machine unlearning experiments. In fact, how exactly it is removing bias in the model? Could you elaborate?

5. Doubts about Resolving Confusion -  I would like to know more about how exactly the confusion is resolved through unlearning?

**Limitations:**

1. The experiments are performed only on CIFAR-10 which is pretty small-scale given the authors mention one of their motivation to design this method is to address the shortcomings of the existing Unlearning methods that work only on small datasets.

2. Several existing methods have already shown results on CIFAR-100 [1,2,3]. The proposed method could outperform some of these methods in 10-class dataset. It would be interesting to see how well the methods perform on 100-class dataset.

3. Sequential Unlearning? – A common use case of Machine Unlearning would require repeated unlearning over the period of time. The experiments on repeated unlearning  is missing in the paper.

---

> ### Author Rebuttal · Authors · 2023-08-08
>
> We thank Reviewer Xo5D for their comments and effort.
>
> **RE: the similarity with Bad-T**
> Thank you for bringing up this important topic. Please see our answer in the general response, above.
>
> **RE: “Wouldn’t that give better robustness against MIAs?”**
> No, that goes against key findings (cf. Carlini et al. for insights on MIAs). To summarize: a model is robust to MIAs if the attacker can’t tell apart training examples from held-out examples (e.g. by inspecting loss values/ confidences). For unlearning the goal is that the forget examples “look" like unseen examples. Having the forget set losses/confidences be similar to those that would come from a random model is thus *not* a good solution, unless the unseen (test) examples’ losses/confidences also looked like they come from a random model (which Bad-T does not encourage and generally is certainly not true). So this approach may lead to increased MIA vulnerability.
>
> On the other hand, SCRUB’s objective for forget examples is to move away from the original model in a controlled manner; via our carefully-crafted min-max steps and rewinding procedure. So SCRUB intuitively may be more successful than Bad-T at driving the unlearned model closer to a region of the desired “retrain-from-scratch” oracle (which again, is very unlikely to be close to random weights). We will add text to explain this better.
>
> **RE: "The proposed method ... prone to being unstable"**
> NB: we developed a practical recipe for optimization that works very well in practice. SCRUB’s very strong performance across a very wide range of scenarios that we consider is a testament to the success of that recipe. On the other hand, we find that Bad-T performs poorly.
>
> **RE: “Bad-T is significantly faster”**
> Bad-T is faster. But it is running for only 1 iteration (as recommended in the Bad-T paper). And when running it for longer it did not further improve! So, it’s worth considering that Bad-T is fast because it doesn’t keep improving with more iterations. SCRUB strikes a great balance, being faster than most, being consistently a top performer across all metrics, applications, datasets, and architectures,  *unlike Bad-T*. Please see also the results from new experiments addressing limitation 2 substantiating the above.
>
> **RE: “Inadequate Literature Study”**
> Reviewer Xo5D is correct: some previous works have moved towards ‘unboundedness’. But SCRUB is the only one shown to be ‘unbound’ in the application domain. We specifically identify that different unlearning applications require different performance metrics for forget quality. The SCRUB/SCRUB+R framework is the only one that is consistently a top performer for all metrics of forget quality for different applications while not sacrificing retain- and test-set accuracy. This is the essential feature of ‘unboundedness’ contributed by this paper and no previous work has achieved this! We thank the reviewer for raising this and plan to make the necessary edits in the Introduction sections to explain this better.
>
> **RE: the rewinding procedure and early stopping**
> There is probably a misunderstanding: SCRUB+R is *fundamentally different to  “just early stopping”*. Early stopping refers to stopping early based on a criterion computed during training (e.g. validation accuracy). SCRUB+R, does not do this! As explained in Section 3.2, it first *must train “all the way”*, until  maximal error on the forget set is obtained. Then, based on that, it determines which previous checkpoint to use (“rewinding”). In addition, SCRUB+R uses an especially-selected subset of the validation set. Overall, our rewinding is not standard. It is a dedicated novel procedure, designed specifically to obtain an appropriate reference point for the forget error and avoid vulnerability to MIAs. Both the mechanism and underlying design principles are far from “just early stopping”.
>
> **RE: the ZRF metric**
> Please note that we don’t “focus just on the Df and Dr accuracies”. We actually do additionally measure generalization. And we are doing so using the most commonly-accepted way of doing so: the accuracy on a test set. In fact, we’re surprised and concerned that test accuracy isn’t reported in the Bad-T paper. The ZRF metric sounds interesting but is definitely a non-standard way to measure generalizability - albeit an interesting idea for future research.
>
> **RE: using CIFAR-100 dataset**
> Indeed we ended up using smaller datasets. This was driven by the fact that several of the competing unlearning algorithms had very poor performance for larger datasets. However, Reviewer Xo5D brings up an interesting issue here.  To address this, we performed experiments in CIFAR-100, using a larger model, namely VGG+BN, with ~138M parameters. Our results in Table 1 to Table 3 (see general response/rebuttal) show that Bad-T fails (one more time) to reconcile model utility with good forget quality. And, (partially) addressing also weakness 1.d. Specifically, Bad-T performs really poorly for retain-set errors and test errors (many times larger than the error of retrain-from-scratch and SCRUB).
>
> **RE: RB and RC experiments**
>
> RB: We consider both class (forgetting a whole class) and selective (forgetting a subset of a class’ examples) unlearning. We assume users/owners of the model have identified the examples producing the biases in an operational model, and place them in the forget set. If unlearning is successful, the biases will have been removed as the effect of those examples are removed from the model.
>
> RC: a subset of the training set is mislabelled and this is discovered after the original model is trained. We place all and only the mislabelled examples in the forget set and run unlearning. If successful, the resulting model will no longer make mistakes due to its past confusion.
>
> In each application, we use the proper metrics to measure forget quality as appropriate.
>
> References:
> - [Carlini et al] Membership inference attacks from first principles. IEEE S&P 2022

---

> > ### Comment · Reviewer_Xo5D · 2023-08-21
> > **Response to the Rebuttale**
> >
> > - Thank you for a detailed response to my queries and doubts. The additional comparisons with Bad-T and more results with CIFAR-100 are a welcome addition.
> >
> > - One important question remained unanswered about Sequential Unlearning. Does the SCRUB support repeated unlearning requests? and how the performance vary as more and more unlearning requests are made over a previously unlearned model?
> >
> > "...As explained in Section 3.2, it first must train “all the way”, until maximal error on the forget set is obtained. Then, based on that, it determines which previous checkpoint to use (“rewinding”). In addition, SCRUB+R uses an especially-selected subset of the validation set. Overall, our rewinding is not standard. It is a dedicated novel procedure, designed specifically to obtain an appropriate reference point for the forget error and avoid vulnerability to MIAs. Both the mechanism and underlying design principles are far from “just early stopping”
> > - This is not clear in the paper. The authors should change the writing in the paper to convey this more clearly.

---

> > > ### Author Response · Authors · 2023-08-21
> > > **response to reviewer Xo5D**
> > >
> > > Dear reviewer,
> > >
> > > Thank you very much for your response acknowledging that out of the 5/5 questions posed in the original review (and 10/10 separate points in these 5 questions) there remain no unanswered questions.  This is very encouraging for us.
> > >
> > > Regarding your point about the one remaining unanswered limitation with respect to sequential unlearning: Indeed we agree. Sequential unlearning is an interesting issue to study. However, we are strongly against reporting any superficial, simply epidermic results that just touch the surface of any important issue. With this paper, we (hope you now agree that we) have made a definite step forward with a new unlearning model and with detailed comprehensive performance analyses across many different metrics for forget quality, model utility, and application scenarios. We would not want to taint this effort with superficial (perhaps misleading) conclusions on related issues that deserve a separate study in their own right.
> > >
> > > Please allow us to elaborate further:
> > >
> > > 1) First, please note that most of the state-of-the-art baselines we compare against do not carefully study the issue of repeated applications of unlearning. And those that do, for instance the Bad-T paper, considers only CIFAR-10 and only 3 repeated unlearning steps.
> > >
> > >
> > > 2) Sequential unlearning adds several layers of complexity to an already complex and relatively young area of research. Indeed, approaching this issue well requires studying the effects of sequential unlearning on:
> > > * the forget errors, and model utility (retain-set and test-set errors)
> > > * other forget-quality metrics, e.g., impact on our two LiRA metrics. For example, there may be no adverse effects on accuracies, but if using different metrics, such as our LiRA ones, such adverse effects may surface!
> > > * (beyond LiRA) the privacy of not just elements in the forget set, but also of elements in the retain set (ie is the privacy of retained examples sacrificed by unlearning other examples?)
> > > * Various applications and upstream and/or downstream tasks
> > > * Very careful experiment designs: for example, there may likely be: i) Different behaviors due to different datasets, different dataset sizes, different tasks (eg 10-way vs 100-way classification) …
> > > ii) Different behavior due to different forget sets vs retain sets. For instance, the similarity of examples in the forget set versus examples in the retain set is likely to be an important factor.
> > >
> > > Hence, (despite the fact that we agree the results shown in the Bad-T paper and in https://arxiv.org/pdf/2111.08947.pdf are a preliminary first step in the right direction) in our opinion it is obvious that (i) there is not enough space to report results on these issues in the paper and (ii) a compehensive/proper evaluation for sequential unlearning is well beyond the scope of this paper.
> > >
> > > Having said the above, we should emphasize that there is no reason why one could not apply SCRUB/+R sequentially. Additionally, we cannot see any intuition or evidence why one would even consider that our method is more vulnerable to unwanted behavior in such a setting compared to any other method.
> > >
> > > Finally, we will indeed follow your suggestion and be clearer in the section describing SCRUB+R about the differences with early stopping. Thank you for this comment.
> > >
> > > Thank you again for your response! Given the above exchanges, we sincerely hope that you now find the paper to be of publishable quality.

---

> > > > ### Comment · Reviewer_Xo5D · 2023-08-22
> > > > **Response**
> > > >
> > > > I am satisfied with the answer and increasing my rating.

---

> > > > > ### Author Response · Authors · 2023-08-22
> > > > >
> > > > > Thank you very much!

---

### Official Review · Reviewer_QAcd · 2023-07-30

**Soundness:** 3 good
**Presentation:** 4 excellent
**Contribution:** 4 excellent
**Rating:** 8
**Confidence:** 3

**Summary:**

This paper introduces the methodology SCRUB, a teacher-student unlearning algorithm which personalises its forget metrics per application and dataset using. This method is demonstrated under three use cases: removing bias, resolving confusion, and user privacy. The authors support their theoretical claims with empirical experiments for each use case.

**Strengths:**

The motivation of this paper is well-founded, and a major contribution to the field of unlearning. The paper is well-written and clear, with the contribution evident. The notation used throughout the paper is easy to follow. The empirical experiments have been well-chosen and SCRUB outperforms other baselines consistently.

**Weaknesses:**

Some of the experiments seem a little simple (specifically the standard attack for the user privacy setting), but given the larger scope of the paper, this is understandable. If additional experiments could be added in this setting, it would greatly strengthen this specific use case.

**Questions:**

- RB, RC, and UP should be defined again in the introduction
- In Section 4, it would be nice to elaborate on why Bad-T performs poorly compared to SCRUB?


**Limitations:**

Yes

---

> ### Author Rebuttal · Authors · 2023-08-08
>
> We thank Reviewer QAcd for the positive feedback. We are glad to hear that the reviewer found our work to be a major contribution to the field of unlearning. We address QAcd’s comments and questions below.
>
> **RE: RB, RC, and UP should be defined again in the introduction**
>
> Great suggestion, we have updated the intro to include that
>
> **RE: In Section 4, it would be nice to elaborate on why Bad-T performs poorly compared to SCRUB?**
>
> To answer this, let’s first consider the differences between SCRUB and Bad-T (please see also our general response for all reviewers for additional details on their differences.
>  SCRUB and Bad-T share the (only) similarity that they can both be formulated using the teacher-student framework (like many other papers within various fields within machine learning). Within the said framework, however, they have fundamentally different objectives:
> 1) SCRUB considers the original model as the teacher and encourages the student to stay close to the teacher on retain examples and move away from it on forget set examples.
> On the other hand
> 2) Bad-T considers two sets of teachers: the original model is the “good teacher”, and they consider a set of *random weights* as a “bad teacher”. They encourage the student to follow the good teacher for retain examples and the bad teacher for forget examples.
>
> So, the two are substantially different, making it hard to draw any conclusions with confidence about the causes of Bad-T’s poor performance. It is not clear why *moving close to random weights* constitutes a good forgetting algorithm. So, we weren’t surprised by the fact that it’s not a strong-performing approach. An adverse effect of moving close to random weights, for instance, may be that the forget set accuracy becomes much lower than it ideally would have been (note that, ideally, the unlearned model has a forget set accuracy similar to that of re-training from scratch without the forget set; not similar to that of a random model). This may be the cause of Bad-T’s poorer performance on MIAs. In addition, we find empirically that Bad-T (with its formulated training objectivEs) fails badly to properly reconcile the trade-offs between good performance on forget set examples vs examples from the retain set. And its generalization ability (performance on test set) is often very poor.
>
> Please note as well that the Bad-T paper does not compare against state-of-the-art methods for unlearning (they only consider one baseline in their paper, which is an older/outdated approach), so an understanding of its empirical performance remained elusive prior to the results we report here.
>
> We hope the above responses address all key concerns and we are very keen to engage in order to further clarify/address comments and questions!

---

### Author Rebuttal · Authors · 2023-08-08

Dear reviewers,
Thank you very much for your dedication and insightful comments. We have thoroughly gone through your comments, taken your suggestions into deep consideration, and provided comprehensive responses.

We answer below a comment that we believe is of interest to all reviewers. It concerns the similarity between SCRUB/SBRUB+R and the Bad-T baseline.  Also, we present 3 tables displaying results from new experiments we have run to address comments by more than one reviewer. These tables serve as supporting evidence for several of the arguments we make in our rebuttal.  The experiments here are conducted on a larger model, i.e VGG16+BN with ~138M parameters, and a larger dataset (in terms of the number of classes), i.e CIFAR-100 and with a variety of forget set sizes.

We address other details in each reviewer's rebuttal section.

**RE: the similarity with Bad-T**
1. SCRUB and Bad-T are only similar in that they both follow the teacher-student (T-S) framework (like many papers within various fields in machine learning). However, the two are *fundamentally different* in terms of their objectives. Specifically: For forget examples, SCRUB encourages the student to move away from the original model (closer in spirit to gradient ascent ideas like NegGrad and *our improved version of NegGrad that we proposed in this paper*) whereas Bad-T encourages it to move towards a random model, yielding fundamentally different unlearning algorithms.

2. NB: the T-S framework is very widely used in machine learning, for a wide variety of methods, including semi-superivsed learning (Xie et al), self-supervised learning (Grill et al), to name a few applications. By no means would one consider those methods to be the same as each other, or that the existence of one deducts ‘novelty points’ from others, just because they both can be thought of as using some kind of a teacher and some kind of a student.

3. KL-divergence between model outputs is a very generic and popular similarity metric that almost every method formulated in a T-S framework utilizes. The choice of this metric, by no means contributes to the novelty of such papers.

4. Our results show SCRUB and Bad-T to have very different performance profiles -- a more reliable indicator of their differences than the common use of terms like “teacher” and “student” in the 2 papers.

There are also other important differences between SCRUB and Bad-T:

5. As an example, SCRUB includes the task loss as well as the distillation loss, making it not strictly a T-S but rather a hybrid objective: aside from obeying/disobeying the teacher, it also optimizes the task loss w.r.t ground-truth labels on the retain set).

6. Further, SCRUB comes with a novel reminding procedure (SCRUB+R) to be triggered in applications where we want to avoid producing an unlearned model whose error is “too high”; making SCRUB uniquely able to handle different application scenarios. Again, we wish to stress as well that our empirical comparison between SCRUB and Bad-T shows SCRUB to significantly outperform Bad-T.

References
- [Xie et al] Self-training with Noisy Student improves ImageNet classification. CVPR 2020.
- [Grill et al] Bootstrap Your Own Latent A New Approach to Self-Supervised Learning. NeurIPS 2020.

**More supporting results**

Table 1: Results for class-unlearning. Class-0 (400 examples) of Cifar-100 (40k examples). The numbers are averaged over 3 runs ± 95% confidence intervals.

| **Method**    | **test-error**| **retain-error**| **forget-error**  | **loss-MIA** |
|:--------------|:-------------:|:-------------:|:-------------:|:-------------:|
| Original      | 35.84±0.47    | 0.07±0.01     | 0.00±0.00     | 61.67±10.04   |
| Retrain       | 36.92±0.63    | 0.09±0.04     | 100.00±0.00   | 55.67±21.13   |
| NegGrad       | 43.16±1.94    | 6.86±3.20     | 49.50±7.93    | 64.67±11.20   |
| Bad-T@epoch1  | 44.14±1.14    | 8.78±1.29     | 63.83±15.27   | 61.67±12.75   |
| Bad-T@epoch10 | 43.33±0.79    | 7.74±1.99     | 85.33±2.59    | 63.33±17.45   |
| SCRUB+R       | 37.60±0.73    | 0.12±0.06     | 100.00±0.00   | 57.33±17.97   |
| SCRUB         | 38.27±1.91    | 0.78±1.53     | 100.00±0.00   | 48.33±7.59    |


Table 2: Results for selective unlearning. 10 examples from class-0 of Cifar-100 dataset.The numbers are averaged over 10 runs ± 95% confidence intervals

| **Method**    | **test-error**| **retain-error**| **forget-error**  | **loss-MIA** |
|:----------------|:-------------:|:-------------:|:-------------:|:-------------:|
| Original        | 37.46±3.09    | 0.54±1.08     | 0.00±0.00     | 58.00±5.64    |
| Retrain         | 37.54±3.25    | 0.57±1.14     | 19.00±9.80    | 45.00±11.80   |
| NegGrad         | 45.98±4.21    | 11.64±7.45    | 44.00±6.91    | 55.00±14.80   |
| Bad-T@epoch1    | 47.23±3.57    | 13.79±6.37    | 66.00±11.29   | 62.00±18.72   |
| Bad-T@epoch10   | 47.09±3.50    | 13.80±6.07    | 96.00±5.00    | 77.00±10.69   |
| SCRUB+R         | 38.40±2.91    | 0.69±1.28     | 40.00±8.26    | 61.00±10.90   |
| SCRUB           | 38.18±2.96    | 0.48±0.89     | 45.00±8.43    | 56.00±12.25   |


Table 3: Results for selective unlearning. 20 examples from class-0 of Cifar-100 dataset.The numbers are average over 10 runs ± 95% confidence intervals

| **Method**    | **test-error**| **retain-error**| **forget-error**  | **loss-MIA** |
|:-------------|:------------:|:------------:|:------------:|:------------:|
| Original     | 37.46±3.09   | 0.54±1.08    | 0.00±0.00    | 63.00±6.79   |
| Retrain      | 37.74±3.29   | 0.56±1.10    | 19.50±7.44   | 49.00±7.11   |
| NegGrad      | 45.56±4.34   | 11.14±7.79   | 47.00±6.13   | 67.50±8.64   |
| Bad-T@epoch1 | 46.46±3.89   | 12.95±7.26   | 55.50±10.04  | 55.00±13.38  |
| Bad-T@epoch10| 45.88±3.53   | 12.03±5.85   | 92.00±6.35   | 70.00±6.74   |
| SCRUB+R      | 38.39±2.87   | 0.68±1.24    | 23.00±5.39   | 49.50±7.24   |
| SCRUB        | 38.29±2.86   | 0.49±0.88    | 29.00±10.09  | 54.50±11.49  |

---

### Decision · Program_Chairs · 2023-09-21

**Decision:**

Accept (poster)

**Comment:**

The paper is clearly written and the experimental evidence for the improvements of the method is rather solid (together with the new results during rebuttals - which the authors should add to the final version!) and they speak for themselves. All in all, it's a classical "method works better" paper with significant improvements and thorough experimental work that the reviewers appreciated. The authors did a good job in the rebuttal as well.